# Initial Value Problem Enhanced Sampling for Closed-Loop Optimal Control Design with Deep Neural Networks

## Abstract

Closed-loop optimal control design for high-dimensional nonlinear systems has been a long-standing challenge. Traditional methods, such as solving the associated Hamilton-Jacobi-Bellman equation, suffer from the curse of dimensionality. Recent literature proposed a new promising approach based on supervised learning, by leveraging powerful open-loop optimal control solvers to generate training data and neural networks as efficient high-dimensional function approximators to fit the closed-loop optimal control. This approach successfully handles certain high-dimensional optimal control problems but still performs poorly on more challenging problems. One of the crucial reasons for the failure is the so-called distribution mismatch phenomenon brought by the controlled dynamics. In this paper, we investigate this phenomenon and propose the initial value problem enhanced sampling method to mitigate this problem. We theoretically prove that this sampling strategy improves over the vanilla strategy on the classical linear-quadratic regulator by a factor proportional to the total time duration. We further numerically demonstrate that the proposed sampling strategy significantly improves the performance on tested control problems, including the optimal landing problem of a quadrotor and the optimal reaching problem of a 7 DoF manipulator.

## 1 Introduction

Optimal control aims to find a control for a dynamical system over a period of time such that a specified loss function is minimized. Generally speaking, there are two types of optimal controls: open-loop optimal control and closed-loop (feedback) optimal control. Open-loop optimal control deals with the problem with a given initial state, and its solution is a function of time for the specific initial data, independent of the other states of the system. In contrast, closed-loop optimal control aims to find the optimal control policy as a function of the state that gives us optimal control for general initial states.

By the nature of the problem, solving the open-loop control problem is relatively easy and various open-loop control solvers can handle nonlinear problems even when the state lives in high dimensions (Betts, 1998; Rao, 2009). Closed-loop control is much more powerful than open-loop control since it can cope with different initial states, and it is more robust to the disturbance of dynamics. The classical approach to obtaining a closed-loop optimal control function is by solving the associated Hamilton-Jacobi-Bellman (HJB) equation. However, traditional numerical algorithms for HJB equations such as the finite difference method or finite element method face the curse of dimensionality (Bellman, 1957) and hence can not deal with high-dimensional problems.

Since the work Han & E (2016) for stochastic optimal control problems, there have been growing interest on making use of the capacity of neural networks (NNs) in approximating high-dimensional functions to solve the closed-loop optimal control problems (Nakamura-Zimmerer et al., 2021a;b; 2020; Böttcher et al., 2022; E et al., 2022). Generally speaking, there are two categories of methods in this promising direction. One is policy search approach (Han & E, 2016; Ainsworth et al., 2021; Böttcher et al., 2022; Zhao et al., 2022), which directly parameterizes the policy function by neural networks, computes the total cost with various initial points, and minimizes the average total cost. When solving problems with a long time span and high nonlinearity, the corresponding optimization

problems can be extremely hard and may get stuck in local minima (Levine & Koltun, 2014). The other category of methods is based on supervised learning (Nakamura-Zimmerer et al., 2021a;b; 2020; 2022). Combining various techniques for open-loop control, one can solve complex high-dimensional open-loop optimal control problems; see Betts (1998); Rao (2009); Kang et al. (2021) for detailed surveys. Consequently, we can collect optimal trajectories for different initial points as training data, parameterize the control function (or value function) using NNs, and train the NN models to fit the closed-loop optimal controls (or optimal values). This work focuses on the second approach and aims to improve its performance through adaptive sampling.

As demonstrated in Nakamura-Zimmerer et al. (2021b); Zang et al. (2022), NN controllers trained by the vanilla supervised-learning-based approach can perform poorly even when both the training error and test error on collected datasets are fairly small. Some existing works attribute this phenomenon to the fact that the learned controller may deteriorate badly at some difficult initial states even though the error is small in the average sense. Several adaptive sampling methods regarding the initial points are hence proposed (see Section 4 for a detailed discussion). However, these methods all focus on choosing optimal paths according to different initial points and ignore the effect of dynamics. This is an issue since the paths controlled by the NN will deviate from the optimal paths further and further over time due to the accumulation of errors. As shown in Section 6, applying adaptive sampling only on initial points is insufficient to solve challenging problems.

This work is concerned with the so-called *distribution mismatch* phenomenon brought by the dynamics in the supervised-learning-based approach. This phenomenon refers to the fact that the discrepancy between the state distribution of the training data and the state distribution generated by the NN controller typically increases over time and the training data fails to represent the states encountered when the trained NN controller is used. Such phenomenon has also been identified in reinforcement learning (Kakade & Langford, 2002; Long & Han, 2022) and imitation learning (Ross & Bagnell, 2010). To mitigate this phenomenon, we propose the initial value problem (IVP) enhanced sampling method to make the states in the training dataset more closely match the states that the controller reaches. In the IVP enhanced sampling method, we iteratively re-evaluate the states that the NN controller reaches by solving IVPs and recalculate new training data by solving the open-loop control problems starting at these states. Our sampling method is very versatile to be combined with other techniques like a faster open-loop control solver or better neural network structures. The resulting supervised-learning-based approach empowered by the IVP enhanced sampling can be interpreted as an instance of the exploration-labeling-training (ELT) algorithms (Zhang et al., 2018; E et al., 2021) for closed-loop optimal control problems (see Appendix A for more discussions). At a high level, the ELT algorithm proceeds iteratively with the following three steps: (1) exploring the state space and examining which states need to be labeled; (2) solving the control problem to label these states and adding them to the training data; (3) training the machine learning model.

The main contributions of the paper can be summarized as follows. (1) We investigate the distribution mismatch phenomenon brought by the controlled dynamics in the supervised-learning-based approach, which explains the failure of this approach for challenging problems. We propose the IVP enhanced sampling method to update the training data, which significantly alleviates the distribution mismatch problem. (2) We show that the IVP enhanced sampling method can significantly improve the performance of the learned closed-loop controller on a uni-dimensional linear quadratic control problem (theoretically and numerically) and two high-dimensional problems (numerically), the quadrotor landing problem and the reaching problem of a 7-DoF manipulator. (3) We compare the IVP enhanced sampling method with other adaptive sampling methods and show that the IVP enhanced method gives the best performance.

## 2 PRELIMINARY

### 2.1 OPEN-LOOP AND CLOSED-LOOP OPTIMAL CONTROL

We consider the following deterministic controlled dynamical system:

$$\begin{cases} \dot{\boldsymbol{x}}(t) = \boldsymbol{f}(t, \boldsymbol{x}(t), \boldsymbol{u}(t)), \ t \in [t_0, T] \\ \boldsymbol{x}(t_0) = \boldsymbol{x}_0 \end{cases} \tag{1}$$

where $\boldsymbol{x}(t) \in \mathbb{R}^n$ denotes the state, $\boldsymbol{u}(t) \in \mathcal{U} \subset \mathbb{R}^m$ denotes the control with $\mathcal{U}$ being the set of admissible controls, $\boldsymbol{f}\colon [0,T] \times \mathbb{R}^n \times \mathcal{U} \to \mathbb{R}^n$ is a smooth function describing the dynamics, $t_0 \in [0,T]$ denotes the initial time, and $\boldsymbol{x}_0 \in \mathbb{R}^n$ denotes the initial state. Given a fixed $t_0 \in [0,T]$ and $\boldsymbol{x}_0 \in \mathbb{R}^n$, solving the open-loop optimal control problem means to find a control path $\boldsymbol{u}^*\colon [t_0, T] \to \mathcal{U}$ to minimize

$$J(\boldsymbol{u}; t_0, \boldsymbol{x}_0) = \int_{t_0}^{\mathrm{T}} L(t, \boldsymbol{x}(t), \boldsymbol{u}(t))\mathrm{d}t + M(\boldsymbol{x}(T)) \ \text{ s.t. } (\boldsymbol{x}, \boldsymbol{u}) \text{ satisfy the system (1),}$$

where $L\colon [0,T] \times \mathbb{R}^n \times \mathcal{U} \to \mathbb{R}$ and $M\colon \mathbb{R}^n \to \mathbb{R}$ are the running cost and terminal cost, respectively. We use $\boldsymbol{x}^*(t; t_0, \boldsymbol{x}_0)$ and $\boldsymbol{u}^*(t; t_0, \boldsymbol{x}_0)$ to denote the optimal state and control with the specified initial time $t_0$ and initial state $\boldsymbol{x}_0$, which emphasizes the dependence of the open-loop optimal solutions on the initial time and state. We assume the open-loop optimal control problem is well-posed, *i.e.*, the solution always exists and is unique.

In contrast to the open-loop control being a function of time only, closed-loop control is a function of the time-state pair $(t, \boldsymbol{x})$. Given a closed-loop control $\boldsymbol{u}\colon [0,T] \times \mathbb{R}^n \to \mathcal{U}$, we can induce a family of the open-loop controls with all possible initial time-state pairs $(t_0, \boldsymbol{x}_0)$:

$$\boldsymbol{u}(t; t_0, \boldsymbol{x}_0) = \boldsymbol{u}(t, \boldsymbol{x}_{\boldsymbol{u}}(t; t_0, \boldsymbol{x}_0)),$$

where $\boldsymbol{x}_{\boldsymbol{u}}(t; t_0, \boldsymbol{x}_0)$ is defined by the following *initial value problem* (IVP):

$$\mathrm{IVP}(\boldsymbol{x}_0, t_0, T, \boldsymbol{u}) : \begin{cases} \dot{\boldsymbol{x}}_{\boldsymbol{u}}(t; t_0, \boldsymbol{x}_0) = \boldsymbol{f}(t, \boldsymbol{x}_{\boldsymbol{u}}(t; t_0, \boldsymbol{x}_0), \boldsymbol{u}(t, \boldsymbol{x}_{\boldsymbol{u}}(t; t_0, \boldsymbol{x}_0))), \ t \in [t_0, T] \\ \boldsymbol{x}_{\boldsymbol{u}}(t_0; t_0, \boldsymbol{x}_0) = \boldsymbol{x}_0. \end{cases} \quad (2)$$

To ease the notation, we always use the same character to denote the closed-loop control function and the induced family of the open-loop controls. The context of closed-loop or open-loop control can be inferred from the arguments and will not be confusing. It is well known in the classical optimal control theory (see, *e.g.* Liberzon (2011)) that there exists a closed-loop optimal control function $\boldsymbol{u}^*\colon [0,T] \times \mathbb{R}^n \to \mathcal{U}$ such that for any $t_0 \in [0,T]$ and $\boldsymbol{x}_0 \in \mathbb{R}^n$,

$$\boldsymbol{u}^*(t; t_0, \boldsymbol{x}_0) = \boldsymbol{u}^*(t, \boldsymbol{x}^*(t; t_0, \boldsymbol{x}_0)),$$

which means the family of the open-loop optimal controls with all possible initial time-state pairs can be induced from the closed-loop optimal control function. Since IVPs can be easily solved, one can handle the open-loop control problems with all possible initial time-state pairs if a good closed-loop control solution is available. Moreover, the closed-loop control is more robust to dynamic disturbance and model misspecification, and hence it is much more powerful in applications.

In this paper, our goal is to find a near-optimal closed-loop control $\hat{\boldsymbol{u}}$ such that for $\boldsymbol{x}_0 \in X \subset \mathbb{R}^n$ with $X$ being the set of initial states of interest, the associated total cost is near-optimal, *i.e.*,

$$|J(\hat{\boldsymbol{u}}(\,\cdot\,; 0, \boldsymbol{x}_0); 0, \boldsymbol{x}_0) - J(\boldsymbol{u}^*(\,\cdot\,; 0, \boldsymbol{x}_0); 0, \boldsymbol{x}_0)| \text{ is small.}$$

## 2.2 Supervised-learning-based Approach for Closed-loop Optimal Control Problem

Here we briefly explain the idea of the supervised-learning-based approach for the closed-loop optimal control problem. The first step is to generate training data by solving the open-loop optimal control problems with zero initial time and initial states randomly sampled in $X$. Then, the training data is collected by evenly choosing points in every optimal path:

$$\mathcal{D} = \{(t^{i,j}, \boldsymbol{x}^{i,j}), \boldsymbol{u}^{i,j}\}_{1 \leq i \leq M, 1 \leq j \leq N},$$

where $M$ and $N$ are the number of sampled training trajectories and the number of points chosen in each path, respectively. Finally, a function approximator (mostly neural network, as considered in this work) with parameters $\theta$ is trained by solving the following regression problem:

$$\min_{\theta} \frac{1}{MN} \sum_{i=1}^{M} \sum_{j=1}^{N} \|\boldsymbol{u}^{i,j} - \boldsymbol{u}^{\mathbf{NN}}(t^{i,j}, \boldsymbol{x}^{i,j}; \theta)\|^2, \quad (3)$$

and gives the NN controller $\boldsymbol{u}^{\mathbf{NN}}$.

## 3    IVP ENHANCED SAMPLING METHOD

Although the vanilla supervised-learning-based approach can achieve a good performance in certain problems (Nakamura-Zimmerer et al., 2021a), it is observed that its performance on complex problems is not satisfactory (see Nakamura-Zimmerer et al. (2021b); Zang et al. (2022) and examples below). One of the crucial reasons that the vanilla method fails is the distribution mismatch phenomenon. To better illustrate this phenomenon, let $\mu_0$ be the distribution of the initial state of interest and $\boldsymbol{u} : [0, T] \times \mathbb{R}^n \to \mathcal{U}$ be a closed-loop control function. We use $\mu_{\boldsymbol{u}}(t)$ to denote the distribution of $\boldsymbol{x}(t)$ generated by $\boldsymbol{u}$: $\dot{\boldsymbol{x}}(t) = \boldsymbol{f}(t, \boldsymbol{x}(t), \boldsymbol{u}(t, \boldsymbol{x}(t))), \boldsymbol{x}_0 \sim \mu_0$. Note that in the training process (3), the distribution of the state at time $t$ is $\mu_{\boldsymbol{u}^*}(t)$, the state distribution generated by the closed-loop optimal control. On the other hand, when we apply the learned NN controller in the dynamics, the distribution of the input state of $\boldsymbol{u}^{\mathbf{NN}}$ at time $t$ is $\mu_{\boldsymbol{u}^{\mathbf{NN}}}(t)$. The error between state $\boldsymbol{x}$ driven by $\boldsymbol{u}^*$ and $\boldsymbol{u}^{\mathbf{NN}}$ accumulates and makes the discrepancy between $\mu_{\boldsymbol{u}^*}(t)$ and $\mu_{\boldsymbol{u}^{\mathbf{NN}}}(t)$ increases over time. Hence, the training data fails to represent the states encountered in the controlled process, and the error between $\boldsymbol{u}^*$ and $\boldsymbol{u}^{\mathbf{NN}}$ dramatically increases when $t$ is large. See Figures 1 (left) and 2 below for an illustration of this phenomenon.

To overcome this problem, we propose the following IVP enhanced sampling method. The key idea is to improve the quality of the NN controller iteratively by enlarging the training dataset with the states seen by the NN controller at previous times. Given predesigned (not necessarily even-spaced) temporal grid points $0 = t_0 < t_1 < \cdots < t_K = T$, we first generate a training dataset $S_0$ by solving open-loop optimal control problems on the time interval $[0, T]$ starting from points in $X_0$, a set of initial points sampled from $\mu_0$, and train the initial model $\hat{\boldsymbol{u}}_0$. Under the control of $\hat{\boldsymbol{u}}_0$, the generated trajectory deviates more and more from the optimal trajectory. So we stop at time $t_1$, $i.e.$, compute the IVPs using $\hat{\boldsymbol{u}}_0$ as the closed-loop control and points in $X_0$ as the initial points on the time interval $[0, t_1]$, and then on the interval $[t_1, T]$ solve new optimal paths that start at the endpoints of the previous IVPs. The new training dataset $S_1$ is then composed of new data (between $t_1$ and T) and the data before time $t_1$ in the dataset $S_0$, and we train a new model $\hat{\boldsymbol{u}}_1$ using $S_1$. We repeat this process to the predesigned temporal grid points $t_2, t_3, \cdots$ until end up with $T$. In other words, in each iteration, the adaptively sampled data replaces the corresponding data (defined on the same time interval) in the training dataset (the size of training data remains the same). The whole process can be formulated as Algorithm 1, and we refer to Figure 1 for an illustration of the algorithm's mechanism. We call this method IVP enhanced sampling method because the initial points of the open-loop optimal control problems are sampled by solving the IVP with the up-to-date NN controller. It is worth noting that the later iterations require less effort in labeling data as the trajectories are shorter and thus easier to solve.

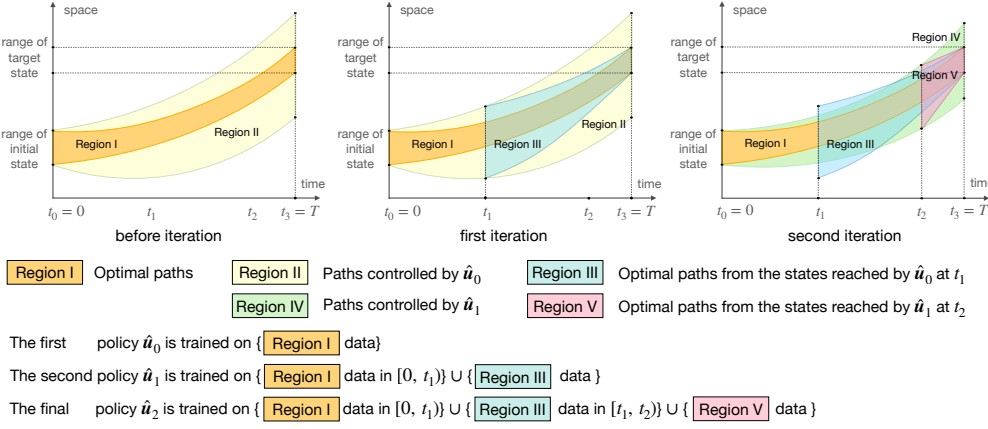

Figure 1: An illustration of how the proposed IVP enhanced sampling (Algorithm 1) works when there are two intermediate temporal grid points $t_1$ and $t_2$.

It is worthwhile mentioning that the IVP enhanced sampling method is versatile enough to combine other improvements for closed-loop optimal control problems, such as efficient open-loop control problem solvers (Kang et al., 2021; Zang et al., 2022) or specialized neural network structures (Nakamura-Zimmerer et al., 2020; 2021b; 2022). One design choice regarding the network structure

---

**Algorithm 1** IVP enhanced sampling method for closed-loop optimal control design

---

1: **Input:** Initial distribution $\mu_0$, number of time points $K$, temporal grid points $0 = t_0 < t_1 < \cdots < t_K = T$, time step $\delta$, number of initial points $N$.
2: **Initialize:** $S_{-1} = \emptyset$, $\hat{\boldsymbol{u}}_{-1}(t, \boldsymbol{x}) = 0$.
3: Independently sample $N$ initial points from $\mu_0$ to get an initial point set $X_0$.
4: **for** $i = 0, 1, \cdots, K - 1$ **do**
5:      For any $\boldsymbol{x}_0 \in X_0$, compute IVP $(\boldsymbol{x}_0, 0, t_i, \hat{\boldsymbol{u}}_{i-1})$ according to (2).        ▷*Exploration*
6:      Set $X_i = \{\boldsymbol{x}_{\hat{\boldsymbol{u}}_{i-1}}(t_i; 0, x_0) : x_0 \in X_0\}$.
7:      For any $\boldsymbol{x}_i \in X_i$, call the open-loop optimal control solver to obtain $\boldsymbol{x}^*(t; t_i, \boldsymbol{x}_i)$ and $\boldsymbol{u}^*(t; t_i, \boldsymbol{x}_i)$ for $t \in [t_i, T]$.        ▷*Labeling*
8:      Set $\hat{S}_i = \{(t, \boldsymbol{x}^*(t; t_i, \boldsymbol{x}_i), \boldsymbol{u}^*(t; t_i, \boldsymbol{x}_i)) : \boldsymbol{x}_i \in X_i, t \in [t_i, T], (t - t_0)/\delta \in \mathbb{N}\}$.
9:      Set $S_i = \hat{S}_i \bigcup \{(t, \boldsymbol{x}, \boldsymbol{u}) : t < t_i, (t, \boldsymbol{x}, \boldsymbol{u}) \in S_{i-1}\}$.
10:     Train $\hat{\boldsymbol{u}}_i$ with dataset $S_i$.        ▷*Training*
11: **end for**
12: **Output:** $\hat{\boldsymbol{u}}_{K-1}$.

---

in the IVP enhanced sampling method is whether to share the same network among different time intervals. We choose to use the same network for all the time intervals in the following numerical examples, but the opposite choice is also feasible.

## 4    COMPARISON WITH OTHER ADAPTIVE SAMPLING METHODS

In this section, we review existing literature on adaptive sampling methods for the closed-loop optimal control problem.

We start with the methods in imitation learning (Hussein et al., 2017), which aim to learn the expert's control function. Our task can be viewed as an imitation problem if we take the optimal control function as the expert's control. With the same argument, we know the distribution mismatch phenomenon also exists therein. However, there is a key difference regarding the mechanism of data generation between the two settings: in imitation learning, it is often assumed that one can easily access the expert's behavior at every time-state pair while in the optimal control problem, it is much more computationally expensive to access since one must solve an open-loop optimal control problem. This difference affects algorithm design fundamentally. Take the forward training algorithm (Ross & Bagnell, 2010), a popular method in imitation learning for mitigating distribution mismatch, as an example. To apply it to the closed-loop optimal control problem, we first need to consider a discrete-time version of the problem with a sufficiently fine time grid: $0 = t_0 < t_1 < \cdots < t_{K'} = T$. At each time step $t_i$, we learn a policy function $\bar{\boldsymbol{u}}^i : \mathbb{R}^n \to \mathcal{U}$ where the state $x$ in the training data are generated by sequentially applying $\bar{\boldsymbol{u}}^0, \ldots, \bar{\boldsymbol{u}}^{i-1}$ and the labels are generated by solving the open-loop optimal solutions with $(t_i, x)$ as the initial time-state pair. Hence, the open-loop control solver is called with numbers proportionally to the discretized time steps $K'$, and only the first value on each optimal control path is used for learning. In contrast, in Algorithm 1, we can use much more values over the optimal control paths in learning, which allows its temporal grid for sampling to be much coarser than the grid in the forward training algorithm, and the total cost of solving open-loop optimal control problems is much lower.

Another popular method in imitation learning is DAGGER (Dataset Aggregation) (Ross et al., 2011), which can also be applied to help sampling in the closed-loop optimal control problem. In DAGGER, in order to improve the current closed-loop controller $\hat{\boldsymbol{u}}$, one solves IVPs using $\hat{\boldsymbol{u}}$ over $[0, T]$ starting from various initial states and collect the states on a time grid $0 < t_1 < \cdots < t_{K-1} < T$. The open-loop control problems are then solved with all the collected time-state pairs as the initial time-state pair, and all the corresponding optimal solutions are used to construct a dataset for learning a new controller. The process can be repeated until a good controller is obtained. The time-state selection in DAGGER is also related to the distribution mismatch phenomenon, but somehow different from the IVP enhanced sampling. Take the data collection using the controller $\hat{\boldsymbol{u}}_1$ in the first iterative step for example. The IVP enhanced sampling focuses on the states at the time grid $t_1$ while DAGGER collects states at all the time grids. If $\hat{\boldsymbol{u}}_1$ is still far from optimal, the data collected at later time grids may be irrelevant to or even mislead training due to error accumulation in states.

In Appendix G, we reports more theoretical and numerical comparison between DAGGER and IVP enhanced sampling, which indicates DAGGER performs less satisfactorily.

Except for the forward training algorithm and DAGGER, there are other adaptive sampling methods for the closed-loop optimal control problems. Nakamura-Zimmerer et al. (2021a) propose an adaptive sampling method that prefers to choose the initial points with large gradients of the value function as the value function tends to be steep and hard to learn around these points. Landry et al. (2021) propose to sample the initial points on which the errors between predicted values from the NN and optimal values are large. These two adaptive sampling methods both focus on finding points that are not learned well but ignore the influence of the accumulation of the distribution mismatch over time brought by controlled dynamics. We will show in Section 6 that the IVP enhanced sampling method can outperform such sampling methods.

## 5 THEORETICAL ANALYSIS ON AN LQR EXAMPLE

In this section, we analyze the superiority of the IVP sampling method by considering the following uni-dimensional linear quadratic regulator (LQR) problem:

$$\min_{x(t),u(t)} \frac{1}{T} \int_{t_0}^{T} |u(t)|^2 \mathrm{d}t + |x(T)|^2$$

$$\text{s.t. } \dot{x}(t) = u(t),\, t \in [t_0, T], \quad x(t_0) = x_0,$$

where $T$ is a positive integer, $t_0 \in [0, T]$ and $x_0 \in \mathbb{R}$. Classical theory on linear quadratic control (see, *e.g.* Sontag (2013)) gives the following explicit linear form of the optimal controls:

$$\begin{cases} u^*(t; t_0, x_0) = -\frac{T}{T(T-t_0)+1} x_0, & \text{(open-loop optimal control)} \\ u^*(t, x) = -\frac{T}{T(T-t)+1} x. & \text{(closed-loop optimal control)} \end{cases}$$

We consider the following two models to approximate the closed-loop optimal control function with parameter $\theta$:

Model 1:    $u_\theta(t, x) = -\frac{T}{T(T-t)+1} x + b(t)$, where $\theta = \{\theta_t\}_{0 \le t \le T} = \{b(t)\}_{0 \le t \le T}$.    (4)

Model 2:    $u_\theta(t, x) = a(t)x + b(t)$, where $\theta = \{\theta_t\}_{0 \le t \le T} = \{(a(t), b(t))\}_{0 \le t \le T}$.    (5)

Since there will be no error in learning a linear model when the data is exact, to mimic the errors encountered when learning neural networks, throughout this section, we assume the data has certain noise. To be precise, for any $t_0 \in [0, T]$ and $x_0 \in \mathbb{R}$, the open-loop optimal control solver gives the following approximated optimal path:

$$\begin{cases} \hat{u}(t; t_0, x_0) = -\frac{T}{T(T-t_0)+1} x_0 + \epsilon Z, \\ \hat{x}(t; t_0, x_0) = x_0 + \int_{t_0}^{t} \hat{u}(t; t_0, x_0) \mathrm{d}t = \frac{T(T-t)+1}{T(T-t_0)+1} x_0 + (t - t_0)\epsilon Z, \end{cases}$$

where $\epsilon > 0$ is a small positive number to indicate the scale of the error and $Z$ is a normal random variable whose mean is $m$ and variance is $\sigma^2$. In other words, the obtained open-loop control is still constant in each path, just like the optimal open-loop control, but perturbed by a random constant. The random variables in different approximated optimal paths starting from different $t_0$ or $x_0$ are assumed to be independent.

We compare the vanilla supervised-learning-based method and IVP enhanced sampling method theoretically for the first model and numerically for the second model (in Appendix B). In the vanilla method, we randomly sample $NT$ initial points from a standard normal distribution and use corresponding optimal paths to learn the controller. In the IVP enhanced sampling method, we randomly sample $N$ initial points from a standard normal distribution, set the temporal grid points for sampling as $0 < 1 < \cdots < T - 1 < T$, and perform Algorithm 1. In both methods, the open-loop optimal control solver is called $NT$ times in total.

Theorem 1 compares the performance of the vanilla method and the IVP enhanced sampling method under Model 1 (4) . The more detailed statement and proof can be found in Appendix B. This theorem shows that both the distribution difference and performance difference with respect to the optimal solution for the vanilla method will increase when $T$ increases, while they are always constantly

bounded for the IVP enhanced sampling method. Therefore, compared to the vanilla method, the IVP enhanced sampling method mitigates the distribution mismatch phenomenon and significantly improves the performance when $T$ is large.

**Theorem 1.** *Under Model 1 (4), let $u_o$, $u_v$ and $u_a$ be the optimal controller, the controller learned by the vanilla method, and the controller learned by the IVP enhanced sampling method, respectively. Define IVPs: $\dot{x}_s(t) = u_s(t) = u_s(t, x_s(t)), x_s(0) = x_{init}, 0 \le t \le T, \quad s \in \{o, v, a\}$.*

1. *If $x_{init}$ is a random variable following a standard normal distribution, which is independent of the initial points and noises in the training process. Let $\{\hat{x}_v^j(t)\}_{j=1}^{NT}$ and $\{\hat{x}_a^j(t)\}_{j=1}^{N}$ be the state variables in the training data of the vanilla method and the last iteration of the IVP enhanced sampling method. Then, $\hat{x}_v^j(t)$, $\hat{x}_a^j(t)$, $x_v(t)$ and $x_a(t)$ are normal random variables, $\mathbb{E}\hat{x}_v^j(t) = \mathbb{E}x_v(t), \mathbb{E}\hat{x}_a^j(t) = \mathbb{E}x_a(t)$ and*

$$|\mathbb{E}|\hat{x}_v^j(t)|^2 - \mathbb{E}|x_v(t)|^2| = (1 - \frac{1}{NT})\epsilon^2 t^2, \quad |\mathbb{E}|\hat{x}_a^j(t)|^2 - \mathbb{E}|x_a(t)|^2| \le \epsilon^2.$$

2. *If $x_{init}$ is a fixed initial point, define the total cost*

$$J_s = \frac{1}{T}\int_0^T |u_s(t)|^2 \mathrm{d}t + |x_s(T)|^2, \quad s \in \{o, v, a\}.$$

*Then, $\mathbb{E}J_v - J_o = (T^2 + 1)(m^2 + \frac{\sigma^2}{NT})\epsilon^2, \quad \mathbb{E}J_a - J_o \le 3(m^2 + \frac{\sigma^2}{N})\epsilon^2.$*

## 6 THE OPTIMAL LANDING PROBLEM OF QUADROTOR

In this section, we test the IVP enhanced sampling method on the optimal landing problem of a quadrotor. We consider the full quadrotor dynamic model with 12-dimensional state variable and 4-dimensional control variable (Bouabdallah et al., 2004; Madani & Benallegue, 2006; Mahony et al., 2012). We aim to find optimal landing paths from some initial states $\boldsymbol{x}_0$ to a target state $\boldsymbol{x}_T = 0$ with minimum control efforts during a fixed time duration $T = 16$. The open-loop optimal solutions are obtained by solving the corresponding two-point boundary value problems with the space-marching technique (Zang et al., 2022). See Appendix C, D and E for more details.

We sample $N = 500$ initial points for generating training data and and use a fully-connected neural network to approximate the optimal control. The temporal grid points on which we do IVP enhanced sampling is $0 < 10 < 14 < 16$. After learning, we use learned models to run the initial value problem at 500 training initial points and show the similarity between paths controlled by the NN controller and their corresponding training data. In Figure 2, the left sub-figure shows the average pointwise distance between data reached by the NN controller and corresponding training data at different times. The right sub-figure shows the maximum mean discrepancy (Borgwardt et al., 2006) between these two datasets using Gaussian kernel $k(x, y) = \exp(-\frac{\|x-y\|^2}{2})$. In both figures, there are jumps at $t = 10$ and $14$ since the NN-controlled path is continuous across time while training data is discontinuous at locations where we do IVP enhanced sampling. It can be seen that without adaptive sampling (after iteration 0), the discrepancy between the states reached by the NN controller and training data is large. With our method, they get closer to each other as the iteration goes.

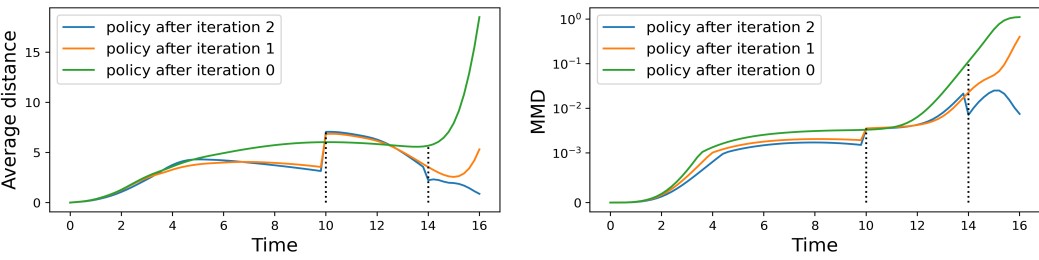

Figure 2: Left: the average pointwise distance between the training data and the data reached by controllers at different times. Right: the maximum mean discrepancy (in the logarithm scale) between the training data and the data reached by controllers at every time using the Gaussian kernel.

Next we check the performance of the learned NN controller. Figure 3 (left) and Figure 3 (middle) show the cumulative distribution function of the ratio between NN-controlled cost and optimal cost on 500 training initial points and 200 test initial points, respectively. Without any adaptive sampling, the performance of model $\hat{u}_0$ is poor even on the training set. In contrast, the NN model is very close to the optimal policy after two rounds of adaptive sampling. More statistics of the results are provided in Appendix E.

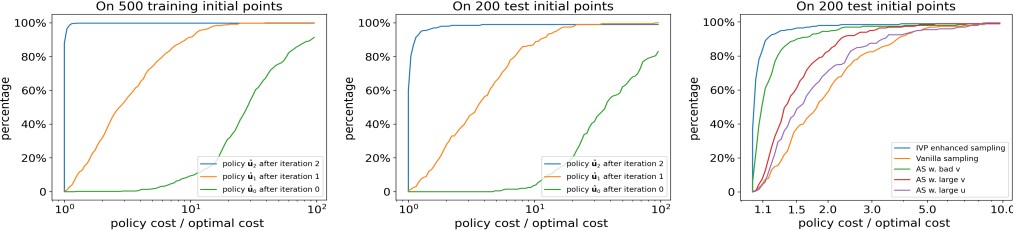

Figure 3: Cumulative distribution function of the cost ratio (with the ideal curve being a straight horizontal segment passing ratio = 1, percentage=100%). Left: results of the IVP enhanced method on 500 training initial points. Middle: results of the IVP enhanced method on 200 test initial points. Right: comparison between different sampling methods.

Then we compare our method with four other methods. As data generation is the most time-consuming part, for the sake of fairness, we keep the number of solving open-loop problems the same (1500) among all the methods (except the last method). The first method is training a model on directly sampled 1500 optimal paths (called *vanilla sampling*). The second method is the adaptive sampling (AS) method proposed by Nakamura-Zimmerer et al. (2021a) that chooses initial points with large gradient norms. This is equivalent to choosing initial points whose optimal control has large norms, and we refer to this method as *AS w. large u*. With a little modification, the third method, *AS w. large v*, is to choose initial points whose total costs are large under the latest NN controller. The last adaptive sampling method, *AS w. bad v*, is a variant of the SEAGuL algorithm (Sample Efficient Adversarially Guided Learning) (Landry et al., 2021). The original SEAGuL algorithm proposes to use a few gradient ascent updates to find initial points with large gaps between the learned values and optimal values. Here we give this method more computational budget to solve more open-loop optimal control problems to find such initial points. The cumulative distribution functions of cost ratios of the above methods are shown in Figure 3 (right), which clearly demonstrate the superiority of the IVP enhanced sampling method. More details about the implementation and results of these methods are provided in Appendix E.

We also test the NN controllers obtained from different sampling methods in the presence of observation noises, considering that the sensors have errors in reality. The detailed results are provided in Appendix E. It is observed that when measurement errors exist, closed-loop controllers are more reliable than the open-loop controller and the one trained by the IVP enhanced sampling method performs best among all the considered methods. Finally, we test 4 different choices of temporal grid points in Algorithm 1, and train networks on the same 500 initial points. The results listed in Appendix E show that our algorithm is robust to the choice of temporal grid points.

## 7 THE OPTIMAL REACHING PROBLEM OF A 7-DOF MANIPULATOR

In this section, we consider the optimal reaching problem on a 7-DoF torque-controlled manipulator, the KUKA LWR iiwa R820 14 (Kuka; Bischoff et al., 2010). See the figure in appendix F for an illustration of this task. Let $x = (q, v) = (q, \dot{q})$ be the state of the system where $q, v \in \mathbb{R}^7$ are joint angles and velocities of the manipulator, respectively. Our goal is to find the optimal torque $u \in \mathcal{U} \subset \mathbb{R}^7$ that drives the manipulator from $x_0 = (q_0, 0)$ to $x_1 = (q_1, 0)$ in $T = 0.8$ seconds and minimizes a quadratic type cost. See Appendix F for details of the problem and the experiment configurations. To obtain training data, we use differential dynamic programming (Jacobson & Mayne, 1970) implemented in the Crocoddy library (Mastalli et al., 2020).

We use the QRNet (Nakamura-Zimmerer et al., 2020; 2021b) as the backbone network in this example (see Appendix F for details) and evaluate networks trained in six different ways: four of them

are trained using Algorithm 1 with different choices of temporal grid points for adaptive sampling and two of them are trained by vanilla sampling method with 300 (*Vanilla300*) and 900 (*Vanilla900*) trajectories separately. All four networks (*AS1–AS4*) trained by the IVP enhanced sampling have initial training data of 100 trajectories and three iterations ($K = 3$), *i.e.*, each of them requires solving the open-loop solution 300 times in total for generating training data. The difference of the four networks lies in that they use different temporal grids ($t_1$ and $t_2$) for enhanced sampling. Each experiment has been independently run five times and we report their average results. We plot the cumulative distribution functions of cost ratios (clipped at 2.0) between the NN-controlled cost and optimal cost in Figure 4 (left). More details and results are provided in Appendix F. We find that adding more data in the vanilla sampling method has very limited effects on improvement while the IVP enhanced sampling greatly improves the performance. Furthermore, such improvement is again robust to different choices of temporal grid points (*AS1–AS4*).

In addition, we also test the performance of the network trained by our adaptive sampling method in the presence of measurement errors. At each simulation timestep, we sample the disturbances uniformly in $[-\sigma, \sigma]^{14}$ for $\sigma = 1e{-}5, 1e{-}4, 1e{-}3$ and add them to the input states of the network. See Figure 4 (right) for the result on the best model trained from *AS1–AS4*. We find that the NN controller performs well at $\sigma = 1e{-}4$, and there are more than $60\%$ of cases on which our controller achieves a ratio less than 2.0 at $\sigma = 1e{-}3$.

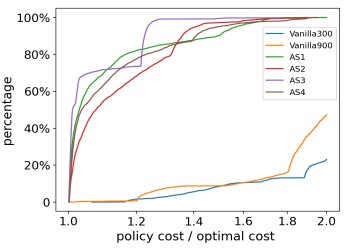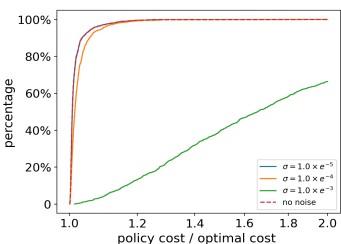

Figure 4: Cumulative distribution functions of cost ratios (with the ideal curve being a straight horizontal segment passing ratio = 1, percentage=$100\%$) under different training schemes (left) and different intensities of measurement noises (right).

## 8 CONCLUSION AND FUTURE WORK

In this work, we propose the IVP enhanced sampling method to overcome the distribution mismatch problem in the supervised-learning-based approaches for the closed-loop optimal control problem. Both theoretical and numerical results show that the IVP enhanced sampling method significantly improves the performance of the learned NN controller and outperforms other adaptive sampling methods. There are a few directions worth exploring in future work. In the IVP enhanced sampling method, one choice we need to make is the temporal grid points for adaptive sampling. We recommend that at each iteration, one can compute the distance between the training data and data reached by the NN controller at different times (see Figure 2 for an example) and choose the time at which the distance starts to increase quickly as the temporal grid for adaptive sampling. We observe that the IVP enhanced sampling method performs well using this strategy. It will be ideal to make this process more systematic. Another direction is to design more effective approaches utilizing the training data. In Algorithm 1 (lines 8–9), at each iteration, we replace parts of the training data with the newly collected data, and hence some optimal labels are thrown away, which are costly to obtain. An alternative choice is to augment data directly, *i.e.*, setting $S_i = \hat{S}_i \bigcup S_{i-1}$ in line 9. Numerically, we observe that this choice gives similar performance to the version used in Algorithm 1, which suggests that so far the dropped data provides little value for training; see Appendix E and F for details. But it is still possible to find smarter ways to utilize them to improve performance. We also need to evaluate the IVP enhanced sampling method for problems with more features like state/control constraints. Furthermore, the IVP enhanced sampling method can be straightforwardly applied to learning general dynamics from multiple trajectories as the controlled system under the optimal closed-loop policy can be viewed as a special dynamical system. It is an interesting direction to investigate its performance in such general settings. Finally, theoretical analysis beyond the LQR setting is also an interesting and important problem.

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

## A  SUPERVISED-LEARNING-BASED APPROACH THROUGH THE LENS OF ELT ALGORITHM

As pointed out in the introduction, the supervised-learning-based approach empowered by the adaptive sampling can be interpreted as an instance of the exploration-labeling-training (ELT) algorithms (Zhang et al., 2018; E et al., 2021) for closed-loop optimal control problems. Through the lens of the ELT algorithm, there are at least three aspects to improve the efficiency of the supervised-learning-based approach for the closed-loop optimal control problem:

- Use the adaptive sampling method. Adaptive sampling methods aim to sequentially choose the time-state pairs based on previous results to improve the performance of the NN controller. This corresponds to the first step in the ELT algorithm and is the main focus of this work.

- Improve the efficiency of data generation, *i.e.*, solving the open-loop optimal control problems. Although the open-loop optimal control problem is much easier than the closed-loop optimal control problem, its time cost cannot be neglected and the efficiency varies significantly with different methods. This corresponds to the second step in the ELT algorithm and we refer to Kang et al. (2021) for a detailed survey.

- Improve the learning process of the neural networks. This corresponds to the third step in the ELT algorithm. The recent works Nakamura-Zimmerer et al. (2020; 2021b; 2022) focus on the structure of the neural networks and design a special ansatz such that the NN controller is close to the linear quadratic controller around the equilibrium point to improve the stability of the NN controller.

## B  DETAILED ANALYSIS OF THE LQR EXAMPLE

In this section, we give the detailed settings for the comparison in Section 5 and the detailed statement and proof of Theorem 1. Through this section, all symbols having a hat are open-loop optimal paths sampled for training, *e.g.* $\hat{u}_j, \hat{x}_j, \hat{u}_j^i, \hat{x}_j^i$. Let $\tilde{x}$ denote a single state instead of a state trajectory. The clean symbol $x$ without hat or tilde is the IVP solution generated by specific controllers which are specified in the subscript; *e.g.* $x_o, x_v, x_a$ are trajectories generated by $u_o, u_v, u_a$ which are optimal, vanilla, and IVP enhanced controllers, respectively. The positive integer $j$ in the subscript always denotes the index of the optimal path. Symbols with superscript $i$ are related to the $i$-th iteration of the IVP enhanced sampling method.

For the vanilla method, we first randomly sample $NT$ initial states $\{\tilde{x}_j\}_{j=1}^{NT}$ from a standard normal distribution where $N$ is a positive integer (recalling $T$ is a positive integer). Then $NT$ approximated optimal paths are collected starting at $t_0 = 0$:

$$\hat{u}_j(t) = -\frac{T}{T^2+1}\tilde{x}_j + \epsilon Z_j, \quad \hat{x}_j(t) = \frac{T(T-t)+1}{T^2+1}\tilde{x}_j + \epsilon t Z_j, \tag{6}$$

where $\{Z_j\}_{j=1}^{NT}$ are *i.i.d.* normal random variables with mean $m$ and variance $\sigma^2$, and independent of initial states. Finally, the parameters $\theta$ are learned by solving the following least square problems:

$$\min_\theta \int_0^T \sum_{j=1}^{NT} |\hat{u}_j(t) - u_\theta(t, \hat{x}_j(t))|^2 \mathrm{d}t. \tag{7}$$

Optimizing $\theta_t$ independently for each $t$, we have

$$\theta_t = \arg\min_b \sum_{j=1}^{NT} |\hat{u}_j(t) + \frac{T}{T(T-t)+1}\hat{x}_j(t) - b|^2 \text{ or } \theta_t = \arg\min_{(a,b)} \sum_{j=1}^{NT} |\hat{u}_j(t) - a\hat{x}_j(t) - b|^2$$

for the first and second models, respectively. We will use $u_v$ to denote the closed-loop controller determined in this way.

For the IVP enhanced sampling method, we choose $K = T$ and the temporal grid points $t_i = i$ for $0 \le i \le K$. We first sample $N$ initial points $\{\tilde{x}_j^0\}_{j=1}^N$ from the normal standard distribution, denote the parameters optimized at $i$-th iteration as $\theta^i$ and initialize $\theta^{-1} = 0$. At the $i$-th iteration ($0 \le i \le T-1$), we use $u_{\theta^{i-1}}$ to solve the IVPs on the time horizon $[0, i]$

$$\dot{x}_j^i(t) = u_{\theta^{i-1}}(t, x_j^i(t)), \quad x_j^i(0) = \tilde{x}_j^0, 1 \le j \le N, \tag{8}$$

and collect $\{\tilde{x}_j^i\}_{j=1}^N$ as $\tilde{x}_j^i := x_j^i(i)$. Here we omit the controller subscript $a$ for simplicity, *i.e.* $x_j^i = x_{a,j}^i$. We then compute $N$ approximated optimal paths starting from $\{\tilde{x}_j^i\}_{j=1}^N$ at $t_i = i$:

$$\hat{u}_j^i(t) = -\frac{T}{T(T-i)+1}\tilde{x}_j^i + \epsilon Z_j^i, \quad \hat{x}_j^i(t) = \frac{T(T-t)+1}{T(T-i)+1}\tilde{x}_j^i + (t-i)\epsilon Z_j^i, t \in [i, T] \tag{9}$$

where $\{Z_j^i\}_{0 \le i \le T-1, 1 \le j \le N}$ are *i.i.d.* normal random variables with mean $m$ and variance $\sigma^2$, and independent of $\{\tilde{x}_j^0\}_{j=1}^N$. Note that $\hat{u}_j^i$ and $\hat{x}_j^i$ are only defined in $t \in [i, T]$ (for $i \ge 1$), we then fill their values in interval $[0, i)$ with values from previous iteration,

$$\hat{u}_j^i(t) = \hat{u}_j^{i-1}(t), \quad \hat{x}_j^i(t) = \hat{x}_j^{i-1}(t), \quad t \in [0, i). \tag{10}$$

Finally, we solve the least square problems to determine $\theta^i$:

$$\min_\theta \int_0^T \sum_{j=1}^N |\hat{u}_j^i(t) - u_\theta(t, \hat{x}_j^i(t))|^2 \mathrm{d}t. \tag{11}$$

We will use $u_a$ to denote the closed-loop controller $u_{\theta^{T-1}}$, the closed-loop controller generated in the $(T-1)$-th iteration by the IVP enhanced sampling method. The theorem below gives the performance of $u_v$ and $u_a$ when using Model 1 (4).

**Theorem 1'.** *Under Model 1* (4)*, define the IVPs generated by $u_o = u^*$, $u_v$ and $u_a$ as follows:*

$$\begin{cases} \dot{x}_o(t) = u_o(t) = u_o(t, x_o(t)), & x_o(0) = \tilde{x}_{init}, 0 \le t \le T, \\ \dot{x}_v(t) = u_v(t) = u_v(t, x_v(t)), & x_v(0) = \tilde{x}_{init}, 0 \le t \le T, \\ \dot{x}_a(t) = u_a(t) = u_a(t, x_a(t)), & x_a(0) = \tilde{x}_{init}, 0 \le t \le T. \end{cases}$$

1. *If $\tilde{x}_{init}$ is a random variable following a standard normal distribution, which is independent of the initial points $\{\tilde{x}_j\}_{j=1}^{NT} / \{\tilde{x}_j^0\}_{j=1}^N$ and noises $\{Z_j\}_{j=1}^{NT} / \{Z_j^i\}_{0 \le i \le T-1, 1 \le j \le N}$ in the training process, the state variables $\hat{x}_j(t)$ in the training data and $x_v(t)$ in the IVP from the vanilla method follow normal distributions and satisfy:*

$$\mathbb{E}\hat{x}_j(t) = \mathbb{E}x_v(t), |\mathbb{E}|\hat{x}_j(t)|^2 - \mathbb{E}|x_v(t)|^2| = \sigma^2(1 - \frac{1}{NT})\epsilon^2 t^2. \tag{12}$$

*On the other hand, the state variables $\hat{x}_j^{T-1}$ in the training data and $x_a(t)$ in the IVP from the IVP enhanced sampling method also follow and satisfy:*

$$\mathbb{E}\hat{x}_j^{T-1}(t) = \mathbb{E}x_a(t), |\mathbb{E}|\hat{x}_j^{T-1}(t)|^2 - \mathbb{E}|x_a(t)|^2| = \sigma^2\epsilon^2(t-i)^2(1-\frac{1}{N}) \le \sigma^2\epsilon^2. \quad (13)$$

*2. If $\tilde{x}_{init}$ is a fixed initial point, define the total cost*

$$J_o = \frac{1}{T}\int_0^T |u_o(t)|^2\mathrm{d}t + |x_o(T)|^2,$$

$$J_v = \frac{1}{T}\int_0^T |u_v(t)|^2\mathrm{d}t + |x_v(T)|^2,$$

$$J_a = \frac{1}{T}\int_0^T |u_a(t)|^2\mathrm{d}t + |x_a(T)|^2.$$

*Then,*

$$\mathbb{E}J_v - J_o = (T^2+1)(m^2 + \frac{\sigma^2}{NT})\epsilon^2, \quad (14)$$

$$\mathbb{E}J_a - J_o \le 3(m^2 + \frac{\sigma^2}{N})\epsilon^2. \quad (15)$$

*Proof.* We first give the closed-form expressions of $u_v$ and $u_a$ using Model 1 (4). Recalling $\hat{u}_j(t)$ and $\hat{x}_j(t)$ given in equation (6), we have

$$\hat{u}_j(t) = -\frac{T}{T(T-t)+1}\hat{x}_j(t) + \frac{T^2+1}{T(T-t)+1}\epsilon Z_j, \ 1 \le j \le NT.$$

Therefore, recalling $u_v$ is learned through the least square problem (7), we have

$$u_v(t,x) = -\frac{T}{T(T-t)+1}x + \frac{T^2+1}{T(T-t)+1}\epsilon\bar{Z}_v, \quad (16)$$

where

$$\bar{Z}_v = \frac{1}{NT}\sum_{j=1}^{NT} Z_j.$$

To compute $u_a$, recalling equations (9) and (10), when $0 \le i \le T-1, 1 \le j \le N$ and $t \in [i, i+1)$, we have

$$\hat{u}_j^{T-1}(t) = \hat{u}_j^i(t) = -\frac{T}{T(T-i)+1}\tilde{x}_j^i + \epsilon Z_j^i,$$

$$\hat{x}_j^{T-1}(t) = \hat{x}_j^i(t) = \frac{T(T-t)+1}{T(T-i)+1}\tilde{x}_j^i + (t-i)\epsilon Z_j^i. \quad (17)$$

Therefore,

$$\hat{u}_j^{T-1}(t) = -\frac{T}{T(T-t)+1}\hat{x}_j^{T-1}(t) + \frac{T(T-i)+1}{T(T-t)+1}\epsilon Z_j^i.$$

Hence, recalling $u_a$ is learned through the least square problem (11), we have, when $t \in [i, i+1)$

$$u_a(t,x) = -\frac{T}{T(T-t)+1}x + \frac{T(T-i)+1}{T(T-t)+1}\epsilon\bar{Z}_a^i, \quad (18)$$

where

$$\bar{Z}_a^i = \frac{1}{N}\sum_{j=1}^{N} Z_j^i, \ 0 \le i \le T-1.$$

Equation (18) also holds when $i = T-1$ and $t = T$.

We then compute the starting points $\{\tilde{x}_j^i\}_{0 \le i \le T-1, 1 \le j \le N}$ in the IVP enhanced sampling method. By equation (10), we know that when $1 \le i \le i' \le T-1$ and $0 \le t < t_i$, $\theta_t^i = \theta_t^{i'}$. Together with equation (8), we know that when $0 \le i \le T-2$, $u_{\theta^i}(t, x) = u_{\theta^{T-1}}(t, x) = u_a(t, x)$ for $t \in [i, i+1)$, and $x_j^{i+1}(t) \equiv x_j^i(t)$ for $t \in [0, i]$, which implies $x_j^{i+1}(i) = x_j^i(i) = \tilde{x}_j^i$. Therefore, for $1 \le j \le N$, when $0 \le i \le T-2$, we have

$$\begin{cases} \dot{x}_j^{i+1}(t) & = u_{\theta^i}(t, x_j^{i+1}(t)) = u_a(t, x_j^{i+1}(t)) \\ & = -\dfrac{T}{T(T-t)+1} x_j^{i+1}(t) + \dfrac{T(T-i)+1}{T(T-t)+1} \epsilon \bar{Z}_a^i, \ t \in [i, i+1], \\ x_j^{i+1}(i) & = \tilde{x}_j^i. \end{cases}$$

Solving the above ODE, we get the solution

$$x_j^{i+1}(t) = \frac{T(T-t)+1}{T(T-i)+1} \tilde{x}_j^i + (t-i)\epsilon \bar{Z}_a^i, \ t \in [i, i+1].$$

Hence, by definition, for $0 \le i \le T-2$,

$$\tilde{x}_j^{i+1} = x_j^{i+1}(i+1) = \frac{T(T-i-1)+1}{T(T-i)+1} \tilde{x}_j^i + \epsilon \bar{Z}_a^i.$$

Utilizing the above recursive relationship, we obtain, for $0 \le i \le T-1$[1],

$$\tilde{x}_j^i = \frac{T(T-i)+1}{T^2+1} \tilde{x}_j^0 + \sum_{k=0}^{i-1} \frac{T(T-i)+1}{T(T-k-1)+1} \epsilon \bar{Z}_a^k. \tag{19}$$

Now we are ready to prove the main results of the Theorem. First, for equation (12), using the control (16), we have

$$\dot{x}_v(t) = -\frac{T}{T(T-t)+1} x_v(t) + \frac{T^2+1}{T(T-t)+1} \epsilon \bar{Z}_v, \ x_v(0) = \tilde{x}_{\text{init}}.$$

Solving this ODE gives

$$x_v(t) = \frac{T(T-t)+1}{T^2+1} \tilde{x}_{\text{init}} + \epsilon t \bar{Z}_v. \tag{20}$$

Combining the last equation with the fact that

$$\hat{x}_j(t) = \frac{T(T-t)+1}{T^2+1} \tilde{x}_j + \epsilon t Z_j,$$

$\tilde{x}_j$, $\tilde{x}_{\text{init}}$ and $\{Z_j\}_{j=1}^{NT}$ are independent normal random variables and

$$\tilde{x}_j, \tilde{x}_{\text{init}} \sim \mathcal{N}(0, 1), Z_j \sim \mathcal{N}(m, \sigma^2),$$

we know that $\hat{x}_j(t)$ and $x_v(t)$ are normal random variables with

$$\mathbb{E}\hat{x}_j(t) = \mathbb{E}x_v(t), |\mathbb{E}|\hat{x}_j(t)|^2 - \mathbb{E}|x_v(t)|^2| = \sigma^2(1 - \frac{1}{NT})\epsilon^2 t^2.$$

Next, we prove the equation (13). for $0 \le i \le T-1$ and $t \in [i, i+1)$, using the control (18), we have

$$\dot{x}_a(t) = -\frac{T}{T(T-t)+1} x_a(t) + \frac{T(T-i)+1}{T(T-t)+1} \epsilon \bar{Z}_a^i.$$

Solving the above ODE with the initial condition $x_a(0) = \tilde{x}_{\text{init}}$, we can get the solution

$$x_a(t) = \frac{T(T-t)+1}{T^2+1} \tilde{x}_{\text{init}} + \sum_{k=0}^{i-1} \frac{T(T-t)+1}{T(T-k-1)+1} \epsilon \bar{Z}_a^k + (t-i)\epsilon \bar{Z}_a^i, \tag{21}$$

when $0 \le i \le T-1$ and $t \in [i, i+1)$. The above equation also holds when $i = T-1$ and $t = T$.

---

[1] In this section, we take by convention that summation $\sum_{k=m}^n c_k = 0$ if $m > n$.

On the other hand, combining equations (17) and (19), we know that when $0 \leq i \leq T - 1$ and $t \in [i, i+1)$ or $i = T - 1$ and $t = T$,

$$
\begin{aligned}
\hat{x}_j^{T-1}(t) &= \frac{T(T-t)+1}{T(T-i)+1} \tilde{x}_j^i + (t-i)\epsilon Z_j^i \\
&= \frac{T(T-t)+1}{T^2+1} \tilde{x}_j^0 + \sum_{k=0}^{i-1} \frac{T(T-t)+1}{T(T-k-1)+1} \epsilon \bar{Z}_a^k + (t-i)\epsilon Z_j^i.
\end{aligned}
$$

The above equation also holds when $i = T - 1$ and $t = T$. Combining the last equation with equation (21) and the fact that $\tilde{x}_j^0, \tilde{x}_{\text{init}}$ and $\{Z_j^i\}_{0 \leq i \leq T-1, 1 \leq j \leq N}$ are independent normal random variables and

$$
\tilde{x}_j^0, \tilde{x}_{\text{init}} \sim \mathcal{N}(0, 1), Z_j^i \sim \mathcal{N}(m, \sigma^2),
$$

we know that $\hat{x}_j^{T-1}(t)$ and $x_a(t)$ are normal random variables with

$$
\mathbb{E}\hat{x}_j^{T-1}(t) = \mathbb{E}x_a(t), |\mathbb{E}|\hat{x}_j^{T-1}(t)|^2 - \mathbb{E}|x_a(t)|^2| = \sigma^2 \epsilon^2 (t-i)^2 (1 - \frac{1}{N}) \leq \sigma^2 \epsilon^2.
$$

We then prove equations (14) and (15). First, with the optimal solution

$$
u_o(t) = -\frac{T}{T^2+1} \tilde{x}_{\text{init}}, \quad x_o(T) = \frac{1}{T^2+1} \tilde{x}_{\text{init}},
$$

we have

$$
J_o = \frac{1}{T} \int_0^T \left| \frac{T}{T^2+1} \tilde{x}_{\text{init}} \right|^2 \mathrm{d}t + \left| \frac{1}{T^2+1} \tilde{x}_{\text{init}} \right|^2 = \frac{1}{T^2+1} |\tilde{x}_{\text{init}}|^2.
$$

Recalling equation (20) and plugging (20) into (16), we know that

$$
x_v(T) = \frac{1}{T^2+1} \tilde{x}_{\text{init}} + \epsilon T \bar{Z}_v,
$$

$$
u_v(t) = -\frac{T}{T^2+1} \tilde{x}_{\text{init}} + \epsilon \bar{Z}_v.
$$

Hence,

$$
J_v = \epsilon^2 |\bar{Z}_v|^2 - \frac{2T}{T^2+1} \tilde{x}_{\text{init}} \epsilon \bar{Z}_v + \epsilon^2 T^2 |\bar{Z}_v|^2 + \frac{2T}{T^2+1} \tilde{x}_{\text{init}} \epsilon \bar{Z}_v + \frac{1}{T^2+1} |\tilde{x}_{\text{init}}|^2,
$$

which gives

$$
\mathbb{E}J_v - J_o = (T^2+1)(m^2 + \frac{\sigma^2}{NT})\epsilon^2.
$$

On the other hand, recalling equation (21) and plugging (21) into (18), we know that

$$
x_a(T) = \frac{1}{T^2+1} \tilde{x}_{\text{init}} + \sum_{k=0}^{T-1} \frac{1}{T(T-k-1)+1} \epsilon \bar{Z}_a^k,
$$

$$
u_a(t) = -\frac{T}{T^2+1} \tilde{x}_{\text{init}} - \sum_{k=0}^{i-1} \frac{T}{T(T-k-1)+1} \epsilon \bar{Z}_a^k + \epsilon \bar{Z}_a^i, \ 0 \leq i \leq T - 1, t \in [i, i+1).
$$

To compute the difference between $J_a$ and $J_o$, we first notice that

$$
\mathbb{E}J_a - J_o = \left[ \frac{1}{T} \int_0^T \mathrm{Var}(u_a(t))\mathrm{d}t + \mathrm{Var}(x_a(T)) \right] + \left[ \frac{1}{T} \int_0^T |\mathbb{E}u_a(t)|^2 \mathrm{d}t + |\mathbb{E}x_a(T)|^2 - J_o \right]
$$

$$
:= \mathrm{I}_1 + \mathrm{I}_2.
$$

By the independence of $\{\bar{Z}_a^i\}_{i=0}^{T-1}$, we know that

$$
\begin{aligned}
\mathrm{I}_1 &= \frac{\sigma^2 \epsilon^2}{NT} \sum_{i=0}^{T-1} \sum_{k=0}^{i-1} \frac{T^2}{[T(T-k-1)+1]^2} + \frac{\sigma^2 \epsilon^2}{N} + \sum_{k=0}^{T-1} \frac{1}{[T(T-k-1)+1]^2} \frac{\sigma^2 \epsilon^2}{N} \\
&= \frac{\sigma^2 \epsilon^2}{N} \left[ 1 + \sum_{i=0}^{T-1} \sum_{k=0}^{i-1} \frac{T}{[T(T-k-1)+1]^2} + \sum_{k=0}^{T-1} \frac{1}{[T(T-k-1)+1]^2} \right] \\
&= \frac{\sigma^2 \epsilon^2}{N} \left[ 1 + \sum_{k=0}^{T-2} \sum_{i=k+1}^{T-1} \frac{T}{[T(T-k-1)+1]^2} + \sum_{k=0}^{T-1} \frac{1}{[T(T-k-1)+1]^2} \right] \\
&= \frac{\sigma^2 \epsilon^2}{N} \left[ 1 + \sum_{k=0}^{T-1} \frac{T(T-k-1)+1}{[T(T-k-1)+1]^2} \right] \\
&= \frac{\sigma^2 \epsilon^2}{N} \left[ 1 + \sum_{k=0}^{T-1} \frac{1}{Tk+1} \right] \\
&\leq \frac{3\sigma^2 \epsilon^2}{N}.
\end{aligned}
$$

Meanwhile, noticing that

$$
\mathbb{E} x_a(T) = \frac{1}{T^2+1} \tilde{x}_{\text{init}} + \sum_{k=0}^{T-1} \frac{1}{T(T-k-1)+1} \epsilon m,
$$

$$
\mathbb{E} u_a(t) = -\frac{T}{T^2+1} \tilde{x}_{\text{init}} - \sum_{k=0}^{i-1} \frac{T}{T(T-k-1)+1} \epsilon m + \epsilon m,
$$

it is straightforward to compute that

$$
\mathrm{I}_2 = \frac{2\epsilon m \tilde{x}_{\text{init}}}{T^2+1} \mathrm{I}_3 + \epsilon^2 m^2 (\mathrm{I}_4 + 1),
$$

where

$$
\begin{aligned}
\mathrm{I}_3 &= \sum_{k=0}^{T-1} \frac{1}{T(T-k-1)+1} + \sum_{i=0}^{T-1} \sum_{k=0}^{i-1} \frac{T}{T(T-k-1)+1} - T \\
&= \sum_{k=0}^{T-1} \frac{1}{T(T-k-1)+1} + \sum_{k=0}^{T-1} \sum_{i=k+1}^{T-1} \frac{T}{T(T-k-1)+1} - T \\
&= \sum_{k=0}^{T-1} \frac{T(T-k-1)+1}{T(T-k-1)+1} - T = 0,
\end{aligned}
$$

and

$$
\begin{aligned}
\mathrm{I}_4 &= \left( \sum_{k=0}^{T-1} \frac{1}{T(T-k-1)+1} \right)^2 + T \sum_{i=0}^{T-1} \left( \sum_{k=0}^{i-1} \frac{1}{T(T-k-1)+1} \right)^2 - \sum_{i=0}^{T-1} \sum_{k=0}^{i-1} \frac{2}{T(T-k-1)+1} \\
&= \sum_{k=0}^{T-1} \frac{1+T(T-k-1)}{[1+T(T-k-1)]^2} + 2 \sum_{i=0}^{T-1} \sum_{k=0}^{i-1} \frac{T(T-i-1)+1}{[T(T-k-1)+1][T(T-i-1)+1]} \\
&\qquad\qquad - 2 \sum_{i=0}^{T-1} \sum_{k=0}^{i-1} \frac{1}{T(T-k-1)+1} \\
&= \sum_{k=0}^{T-1} \frac{1}{1+T(T-k-1)} = \sum_{k=0}^{T-1} \frac{1}{Tk+1} \leq 2.
\end{aligned}
$$

Therefore,

$$\mathbb{E}J_a - J_o = \mathrm{I}_1 + \frac{2\epsilon m \tilde{x}_{\mathrm{init}}}{T^2 + 1}\mathrm{I}_3 + \epsilon^2 m^2(\mathrm{I}_4 + 1) \le 3(m^2 + \frac{\sigma^2}{N})\epsilon^2.$$

□

Next we present the numerical results when we use Model 2 (5) to fit the closed-loop optimal control. In the following experiments, we set $\epsilon = 0.1$, $m = 0.1$ and $\sigma^2 = 1$. Figure 5 (left) compares the optimal path $x_o(t)$ with $x_v(t)$ and $x_a(t)$, the IVPs generated by the controllers learned by the vanilla method and IVP enhanced sampling method (all three paths start at $x_{\mathrm{init}} = 1$). In this experiment, we set $T = 30$ and $N = 100$. Figure 5 (middle) shows how the time $t$ influences the differences of the second order moments between the state distribution of the training data and the state distribution of the IVP generated by learned controllers in the vanilla method and the IVP enhanced sampling method. We set the total time $T = 100$ and $N = 100$. Figure 5 (right) compares the performance of the vanilla method and IVP enhanced sampling method on different total times $T$. The performance difference is an empirical estimation of $\mathbb{E}[J_v - J_o]$ and $\mathbb{E}[J_a - J_o]$ when $x_{\mathrm{init}}$ follows a standard normal distribution. In this experiment, for each method, we set $N = 100$ and learn 10 different controllers with different realizations of the training data and calculate the average of the performance difference on 1000 randomly sampled initial points (from a standard normal distribution) and 10 learned controllers.

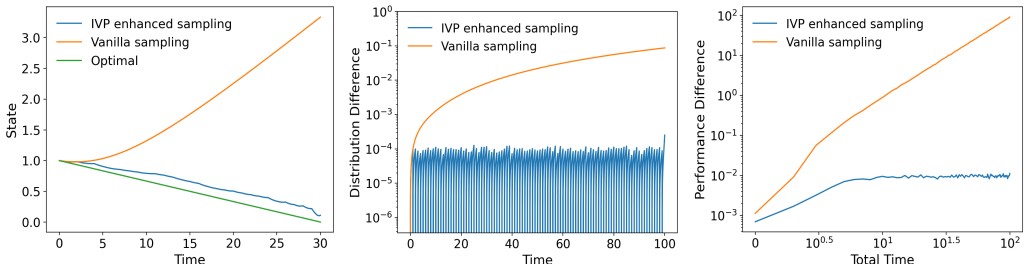

Figure 5: Numerical results on learning Model 2 (5). Left: the optimal path and the paths generated by the vanilla sampling method and the IVP enhanced sampling method. Middle: differences of the second order moments (in the logarithm scale) between the distributions of the training data and the data reached by the controllers at different times. Right: performance differences (in the logarithm scale) of the vanilla sampling method and the IVP enhanced sampling method for different total times (in the logarithm scale).

## C  FULL DYNAMICS OF QUADROTOR

In this section, we introduce the full dynamics of quadrotor (Bouabdallah et al., 2004; Madani & Benallegue, 2006; Mahony et al., 2012) that are considered in Section 6. The state variable of a quadrotor is $\boldsymbol{x} = (\boldsymbol{p}^{\mathrm{T}}, \boldsymbol{v}_b^{\mathrm{T}}, \boldsymbol{\eta}^{\mathrm{T}}, \boldsymbol{w}_b^{\mathrm{T}})^{\mathrm{T}} \in \mathbb{R}^{12}$ where $\boldsymbol{p} = (x, y, z) \in \mathbb{R}^3$ is the position of quadrotor in Earth-fixed coordinates, $\boldsymbol{v}_b \in \mathbb{R}^3$ is the velocity in body-fixed coordinates, $\boldsymbol{\eta} = (\phi, \theta, \psi) \in \mathbb{R}^3$ (roll, pitch, yaw) is the attitude in terms of Euler angles in Earth-fixed coordinates, and $\boldsymbol{w}_b \in \mathbb{R}^3$ is the angular velocity in body-fixed coordinates. Control $\boldsymbol{u} = (s, \tau_x, \tau_y, \tau_z)^{\mathrm{T}} \in \mathbb{R}^4$ is composed of total thrust $s$ and body torques $(\tau_x, \tau_y, \tau_z)$ from the four rotors. Then we can model the quadrotor's dynamics as

$$\begin{cases} \dot{\boldsymbol{p}} = \boldsymbol{R}^{\mathrm{T}}(\boldsymbol{\eta})\boldsymbol{v}_b \\ \dot{\boldsymbol{v}}_b = -\boldsymbol{w}_b \times \boldsymbol{v}_b - \boldsymbol{R}(\boldsymbol{\eta})\boldsymbol{g} + \frac{1}{m}A\boldsymbol{u} \\ \dot{\boldsymbol{\eta}} = \boldsymbol{K}(\boldsymbol{\eta})\boldsymbol{w}_b \\ \dot{\boldsymbol{w}}_b = -\boldsymbol{J}^{-1}\boldsymbol{w}_b \times \boldsymbol{J}\boldsymbol{w}_b + \boldsymbol{J}^{-1}B\boldsymbol{u}, \end{cases}$$

with matrix $A$ and $B$ defined as

$$A = \begin{bmatrix} 0 & 0 & 0 & 0 \\ 0 & 0 & 0 & 0 \\ 1 & 0 & 0 & 0 \end{bmatrix}, \qquad B = \begin{bmatrix} 0 & 1 & 0 & 0 \\ 0 & 0 & 1 & 0 \\ 0 & 0 & 0 & 1 \end{bmatrix}.$$

The constant mass $m$ and inertia matrix $\boldsymbol{J} = \mathrm{diag}(J_x, J_y, J_z)$ are the parameters of the quadrotor, where $J_x$, $J_y$, and $J_z$ are the moments of inertia of the quadrotor in the $x$-axis, $y$-axis, and $z$-axis, respectively. We set $m = 2kg$ and $J_x = J_y = \frac{1}{2}J_z = 1.2416 kg \cdot m^2$ which are the same system parameters as in (Madani & Benallegue, 2006). The constants $\boldsymbol{g} = (0, 0, g)^{\mathrm{T}}$ denote the gravity vector where $g = 9.81 m/s^2$ is the acceleration of gravity on Earth. The direction cosine matrix $\boldsymbol{R}(\boldsymbol{\eta}) \in SO(3)$ represents the transformation from the Earth-fixed coordinates to the body-fixed coordinates:

$$\boldsymbol{R}(\boldsymbol{\eta}) = \begin{bmatrix} \cos\theta\cos\psi & \cos\theta\sin\psi & -\sin\theta \\ \sin\theta\cos\psi\sin\phi - \sin\psi\cos\phi & \sin\theta\sin\psi\sin\phi + \cos\psi\cos\phi & \cos\theta\sin\phi \\ \sin\theta\cos\psi\cos\phi + \sin\psi\sin\phi & \sin\theta\sin\psi\cos\phi - \cos\psi\sin\phi & \cos\theta\cos\phi \end{bmatrix},$$

and the attitude kinematic matrix $\boldsymbol{K}(\boldsymbol{\eta})$ relates the time derivative of the attitude representation with the associated angular rate:

$$\boldsymbol{K}(\boldsymbol{\eta}) = \begin{bmatrix} 1 & \sin\phi\tan\theta & \cos\phi\tan\theta \\ 0 & \cos\phi & -\sin\phi \\ 0 & \sin\phi\sec\theta & \cos\phi\sec\theta \end{bmatrix},$$

Note that in practice the quadrotor is directly controlled by the individual rotor thrusts $\boldsymbol{F} = (F_1, F_2, F_3, F_4)^{\mathrm{T}}$, and we have the relation $\boldsymbol{u} = E\boldsymbol{F}$ with

$$E = \begin{bmatrix} 1 & 1 & 1 & 1 \\ 0 & l & 0 & -l \\ -l & 0 & l & 0 \\ c & -c & c & -c \end{bmatrix},$$

where $l$ is the distance from the rotor to the UAV's center of gravity and $c$ is a constant that relates the rotor angular momentum to the rotor thrust (normal force). So once we obtain the optimal control $\boldsymbol{u}^*$, we are able to get the optimal $\boldsymbol{F}^*$ immediately by the relation $\boldsymbol{F}^* = E^{-1}\boldsymbol{u}^*$.

## D  PMP AND SPACE MARCHING METHOD

In this section we introduce the open-loop optimal problem solver used for solving the optimal landing problem of a quadrotor. The solver is based on Pontryagin's minimum principle (PMP) (Pontryagin, 1987) and space-marching method (Zang et al., 2022). The optimal landing problem is defined as

$$\min_{\boldsymbol{x}, \boldsymbol{u}} \int_0^T L(\boldsymbol{x}(\tau), \boldsymbol{u}(\tau))d\tau + M(\boldsymbol{x}(T)),$$
$$\text{s.t.} \begin{cases} \dot{\boldsymbol{x}}(t) = f(\boldsymbol{x}(t), \boldsymbol{u}(t)), t \in [0, T], \\ \boldsymbol{x}(0) = \boldsymbol{x}_0, \end{cases} \tag{22}$$

where $\boldsymbol{x}(t) : [0, T] \to \mathbb{R}^{12}$ and $\boldsymbol{u}(t) : [0, T] \to \mathbb{R}^4$ denote the state trajectory and control trajectory, respectively, and $f$ is the full dynamics of quadrotor introduced in Appendix C. By PMP, problem (22) can be solved through solving a two-point boundary value problem (TPBVP). Introduce costate variable $\boldsymbol{\lambda} \in \mathbb{R}^{12}$ and Hamiltonian

$$H(\boldsymbol{x}, \boldsymbol{\lambda}, \boldsymbol{u}) = L(\boldsymbol{x}, \boldsymbol{u}) + \boldsymbol{\lambda} \cdot f(\boldsymbol{x}, \boldsymbol{u}).$$

The TPBVP is defined as

$$\begin{cases} \dot{\boldsymbol{x}}(t) = \partial_{\boldsymbol{\lambda}}^T H\left(\boldsymbol{x}(t), \boldsymbol{\lambda}(t), \boldsymbol{u}^*(t)\right), \\ \dot{\boldsymbol{\lambda}}(t) = -\partial_{\boldsymbol{x}}^T H\left(\boldsymbol{x}(t), \boldsymbol{\lambda}(t), \boldsymbol{u}^*(t)\right). \end{cases} \tag{23}$$

We have the boundary conditions:

$$\begin{cases} \boldsymbol{x}(0) = \boldsymbol{x}_0, \\ \boldsymbol{\lambda}(T) = \nabla M\left(\boldsymbol{x}(T)\right), \end{cases}$$

and the optimal control $\boldsymbol{u}^*(t)$ should minimize Hamiltonian at each $t$:

$$\boldsymbol{u}^*(t) = \arg\min_{\boldsymbol{u}} H(\boldsymbol{x}(t), \boldsymbol{\lambda}(t), \boldsymbol{u}). \tag{24}$$

We use *solve_bvp* function of *scipy* (Kierzenka & Shampine, 2001) to solve TPBVP (23)-(24) and set *tolerance* to $10^{-5}$, *max_nodes* to 5000. We note that when the initial state $\boldsymbol{x}_0$ is far from the target state $\boldsymbol{x}_T$, solving the TPBVP directly often fails. Thus we use the space-marching method proposed in Zang et al. (2022). We uniformly select $K$ points in the line segment from $\boldsymbol{x}_T$ to $\boldsymbol{x}_0$, and denote them as $\{\boldsymbol{x}_0^1, \boldsymbol{x}_0^2, \cdots, \boldsymbol{x}_0^K\}$ according to their increasing distances to $\boldsymbol{x}_T$ ($x_0^K = \boldsymbol{x}_0$). These $K$ TPBVPs will be solved in order and at every step we use the previous solution as the initial guess to the current problem.

## E   EXPERIMENT DETAILS AND MORE RESULTS OF THE OPTIMAL LANDING PROBLEM

In this section, we give more details about implementation and numerical results in Section 6. We aim to find the optimal controls to steer the quadrotor from some initial states $\boldsymbol{x}_0$ to a target state $\boldsymbol{x}_T = \boldsymbol{0}$. The distribution of the initial state of interest is the uniform distribution on the set $X = \{x, y \in [-40, 40], z \in [20, 40], v_x, v_y, v_z \in [-1, 1], \theta, \phi \in [-\pi/4, \pi/4], \psi \in [-\pi, \pi]; \boldsymbol{w} = \boldsymbol{0}\}$. We consider a quadratic running cost:

$$L(\boldsymbol{x}, \boldsymbol{u}) = (\boldsymbol{u} - \boldsymbol{u}_d)^{\mathrm{T}} Q_u (\boldsymbol{u} - \boldsymbol{u}_d),$$

where $\boldsymbol{u}_d = (mg, 0, 0, 0)$ represents the reference control that balances with gravity and $Q_u = \text{diag}(1, 1, 1, 1)$ represents the weight matrix characterizing the cost of deviating from the reference control. The terminal cost is

$$M(\boldsymbol{x}) = \boldsymbol{p}^{\mathrm{T}} Q_{pf} \boldsymbol{p} + \boldsymbol{v}^{\mathrm{T}} Q_{vf} \boldsymbol{v} + \boldsymbol{\eta}^{\mathrm{T}} Q_{\eta f} \boldsymbol{\eta} + \boldsymbol{w}^{\mathrm{T}} Q_{wf} \boldsymbol{w} = \boldsymbol{x}^{\mathrm{T}} Q_f \boldsymbol{x}$$

where $Q_{pf} = 5I_3$, $Q_{vf} = 10I_3$, $Q_{\eta f} = 25I_3$, $Q_{wf} = 50I_3$. We set the entries in the terminal cost larger than the running cost as we want to give the endpoint more penalty for deviating from the landing target.

We sample $N = 500$ initial points for training. On every optimal path, we select time-state-action tuples with time step $\delta = 0.2$. Thus the number of training data is always $81 \times 500$ at every iteration. Note that when solving BVPs and IVPs, we use denser time grids to ensure the solution is accurate enough. The neural network models in all quadrotor experiments have the same structure with 13-dimensional input (12 for states and 1 for time) and 4-dimensional output. The networks are fully-connected with 2 hidden layers; each layer has 128 hidden neurons and we use $\tanh$ as the activation function. The inputs are scaled to $(-1, 1)$ where the upper bound and lower bound are the maximum and minimum of the training dataset. Since the activation fucntion is $\tanh$, we adapt Xavier initialization (Glorot & Bengio, 2010) before training. We train the neural network by the Adam (Kingma & Ba, 2015) optimizer with learning rate 0.001, batch size 1000, and 1000 epochs. At every iteration of IVP enhanced sampling method, we train a new neural network from scratch. Our model and training programs are implemented by PyTorch (Paszke et al., 2019).

In the first experiment, we use our IVP enhanced sampling method and choose temporal grid points $0 < 10 < 14 < 16$. More statistics of the ratios between the NN-controlled costs and the optimal costs during three iterations are shown in Table 1. We also test an alternative way to construct dataset during the IVP enhanced sampling as discussed in Section 8, *i.e.*, setting $S_i = \hat{S}_i \bigcup S_{i-1}$ in Algorithm 1 line 9. We denote the final policy obtained in this approach as $\tilde{\boldsymbol{u}}$ and report the statistics of corresponding cost ratios in Table 1 as well. We have to point out that the ratio of $\tilde{\boldsymbol{u}}$ has been clipped at 10.0 as there is a test path with a ratio over 1000. The performance of $\tilde{\boldsymbol{u}}$ is similar to that of $\hat{\boldsymbol{u}}_2$, suggesting that so far the dropped data provides little value for training. Additionally, as we always train networks 1000 epochs and this alternative approach has more training data, the training time of $\tilde{\boldsymbol{u}}$ is 1.5 times that of others.

We also illustrate the trajectories of states controlled by learned policy and optimal policy in Figure 6. The path controlled by $\hat{\boldsymbol{u}}_0$ matches the optimal path at the beginning but deviates around $t = 10$. Then the path controlled by $\hat{\boldsymbol{u}}_1$ fits the optimal path more and deviates around $t = 14$. Finally, the path controlled by $\hat{\boldsymbol{u}}_3$ matches the optimal path till the terminal time. Note that the cost of three controlled paths is 3296.18, 119.91, 6.69, respectively, and the optimal cost is 6.32.

| Policy | Mean | Std | Max | 90% | 75% | Median |
|--------|------|-----|-----|-----|-----|--------|
| $\hat{u}_0$ | 45.52 | 55.73 | 521.98 | 23.53 | 20.32 | 17.32 |
| $\hat{u}_1$ | 4.41 | 4.17 | 39.69 | 2.24 | 2.01 | 1.67 |
| $\hat{u}_2$ | 1.02 | 0.03 | 1.30 | 1.01 | 1.01 | 1.01 |
| $\tilde{u}$ | 1.03 | 0.04 | 1.43 | 1.01 | 1.01 | 1.01 |

(a) on 500 training points

| Policy | Mean | Std | Max | 90% | 75% | Median |
|--------|------|-----|-----|-----|-----|--------|
| $\hat{u}_0$ | 67.90 | 86.14 | 578.05 | 145.82 | 74.13 | 38.21 |
| $\hat{u}_1$ | 5.57 | 7.70 | 91.63 | 11.54 | 6.52 | 3.74 |
| $\hat{u}_2$ | 1.09 | 0.25 | 3.95 | 1.16 | 1.08 | 1.04 |
| $\tilde{u}$ | 1.14 | 0.64 | 10.00 | 1.24 | 1.10 | 1.05 |

(b) On 200 test points

Table 1: The numerical results of the IVP enhanced sampling method on training and test points. The results in Table (a) and (b) correspond to Figure 3 (left) and 3 (middle), respectively. $\hat{u}_0$, $\hat{u}_1$, $\hat{u}_2$ denote the policy after the first, second, and third round of training, respectively. $\tilde{u}$ denotes the final policy obtained from an alternative approach for constructing datasets during the iteration of the IVP enhanced sampling; see discussion in Section 8.

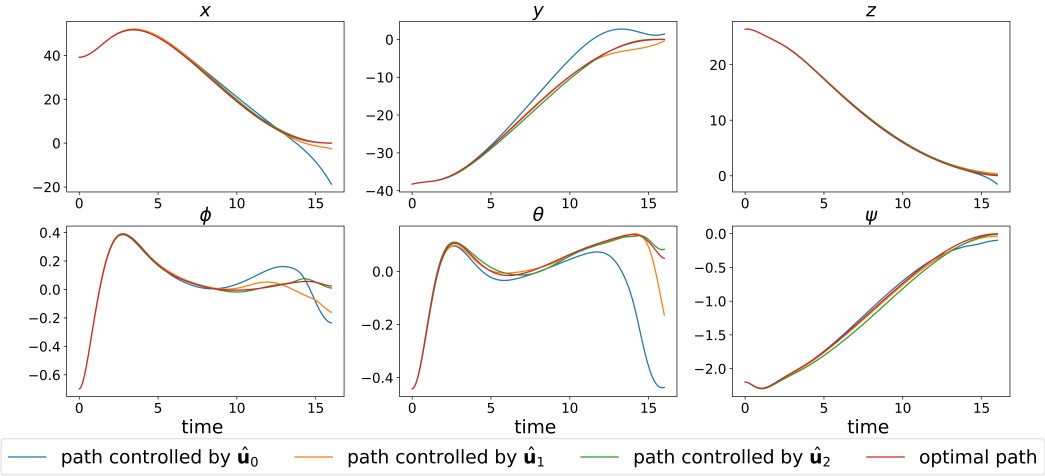

Figure 6: The optimal path and path controlled by learned controllers. We show the 3-dimensional position $p = (x, y, z)$ and 3-dimensional attitude $\eta = (\phi, \theta, \psi)$ in terms of Euler angles in Earth-fixed coordinates.

In the experiments of comparing different methods, the three adaptive sampling methods are processed as follows. The initial network is the same policy as in the IVP enhanced sampling method, *i.e.*, the policy $\hat{u}_0$ after iteration 0 in the IVP enhanced sampling method, which is trained on 500 optimal paths. Then we will sequentially add 400, 300, 300 paths to training data so we use 1500 optimal paths to train the final network (solving 1500 open-loop optimal control problems namely). When sampling new training paths, we will randomly sample 2 initial points, calculate some indices using the latest policy and preserve the preferable one. For *AS w. large u*, we will calculate the norms of the control variables of these two points at time 0 and choose the one with larger norms. For *AS w. large v*, we will solve IVPs to obtain the NN-controlled costs and choose the larger one. For *AS w. bad v*, we will solve 2 TPBVPs corresponding to these two points and choose the one with a large difference between the NN-controlled value and optimal value. Note that in all other methods, the time span of the solved open-loop problem is always $T$. Nevertheless, in our method, the time span of the optimal paths to be solved is getting shorter and shorter along iterations, which means it takes less time to solve. Specifically, in our experiment, it takes about 11.1s to solve an

optimal path whose total time $T - t_0 = 16$ at iteration 0 and takes about 3.1s to solve an optimal path whose total time $T - t_2 = 2$ at iteration 2. More statistics are shown in Table 2.

| Methods | Mean | Std | Max | 90% | 75% | Median |
|---|---|---|---|---|---|---|
| **IVP enhanced sampling** | 1.09 | 0.25 | 3.95 | 1.16 | 1.08 | 1.04 |
| **Vanilla sampling** | 2.27 | 1.39 | 10.11 | 3.93 | 2.54 | 1.86 |
| **AS w. large u** | 2.04 | 1.34 | 8.70 | 3.39 | 2.21 | 1.57 |
| **AS w. large v** | 1.61 | 0.66 | 5.56 | 2.28 | 1.75 | 1.37 |
| **AS w. bad v** | 1.32 | 0.84 | 8.49 | 1.54 | 1.26 | 1.12 |

Table 2: Comparison on different sampling methods.

We further test the performance of NN controllers obtained from different sampling methods in the presence of observation noises, considering that the sensors have errors in reality. During simulation, we add a disturbance $\epsilon$ to the input of the network, where $\epsilon \in \mathbb{R}^{13}$ (including the disturbance of time) is uniformly sampled from $[-\sigma, \sigma]^{13}$. We test $\sigma = 0.01, 0.05, 0.1$ and the numerical results are shown in Figure 7. We also test the performance of the open-loop optimal controller under perturbation where a disturbance $\hat{\epsilon} \in \mathbb{R}$ is added to the input time. Figure 7 shows that when disturbance exists, closed-loop controllers are more reliable than the open-loop controller and the one trained by the IVP enhanced sampling method performs best among all the methods.

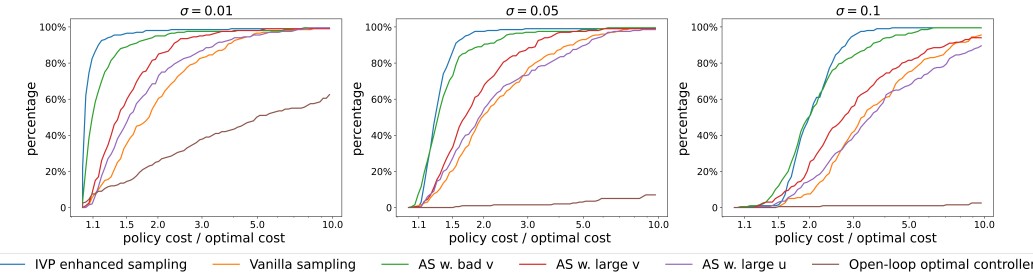

Figure 7: Cumulative distribution function of the cost ratio between NN controlled value and the optimal value under disturbance.

Finally we consider the impact of different choices of temporal grid points in Algorithm 1. We test 4 experiments which are all trained using the same 500 initial points. The results listed in Table 3 show that our algorithm is robust to the choice of temporal grid points.

| # **of iterations** | **Temporal grid points** $(T = 16)$ | | | | | |
|---|---|---|---|---|---|---|
| | 0 | 4 | 8 | 10 | 12 | 14 |
| 2 | 67.90 | | | | | 1.14 |
| 3 | 67.90 | | | 5.57 | | 1.09 |
| 4 | 67.90 | | 37.11 | | 2.34 | 1.18 |
| 5 | 67.90 | 37.66 | 11.13 | | 1.79 | 1.32 |

Table 3: Average cost ratio on 200 test points of every model. The first line means that we take 2 iterations and the corresponding temporal grid points for adaptive sampling are $0 < 14 < 16$. The average ratio of policy cost / optimal cost is 67.90 after iteration 0 and the average ratio after iteration 1 is 1.14. The second line shows the same experiment in Figure 3 (middle).

## F   EXPERIMENT DETAILS AND MORE RESULTS OF THE MANIPULATOR

We can write down the dynamics of the manipulator as,

$$\dot{x} = f(x, u) = (v, a(x, u)),$$

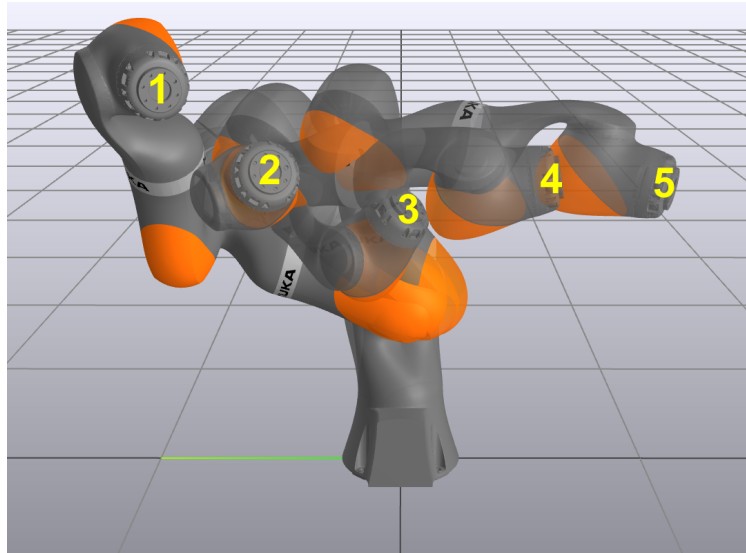

Figure 8: An illustration of the reaching problem of the manipulator. The solid manipulator demonstrates its initial position. We label the end effectors of the five instances of robots by "1,2,3,4,5" to indicator the position of the robot at different times $t_1 = 0.0, t_2 = 0.2, t_3 = 0.4, t_4 = 0.6, t_5 = 0.8$.

where $\boldsymbol{u} \in \mathbb{R}^7$ is the control torque, $\boldsymbol{x} = (\boldsymbol{q}, \boldsymbol{v}) \in \mathbb{R}^{14}$, $\boldsymbol{q} \in \mathbb{R}^7$ is the joint angles, $\boldsymbol{v} = \dot{\boldsymbol{q}} \in \mathbb{R}^7$ is the joint velocities, $\ddot{\boldsymbol{q}} = \boldsymbol{a}(\boldsymbol{x}, \boldsymbol{u}) \in \mathbb{R}^7$ is the acceleration of joint angles.

The acceleration $\boldsymbol{a}$ is given by the (non-linear) forward dynamics

$$M(\boldsymbol{q})\boldsymbol{a} + C(\boldsymbol{q}, \dot{\boldsymbol{q}})\dot{\boldsymbol{q}} + \boldsymbol{g}(\boldsymbol{q}) = \boldsymbol{u}.$$

Here $M(\boldsymbol{q})$ is the generalized inertia matrix, $C(\boldsymbol{q}, \dot{\boldsymbol{q}})\dot{\boldsymbol{q}}$ represents the *centrifugal* forces and *Coriolis* forces, and $\boldsymbol{g}(\boldsymbol{q})$ is the generalized gravity.

The reaching task is to move the manipulator from the initial states near $\boldsymbol{x}_0$ to the terminal states $\boldsymbol{x}_1$. In the experiments, we take $\boldsymbol{x}_0 = (\boldsymbol{q}_0, \boldsymbol{0})$, $\boldsymbol{x}_1 = (\boldsymbol{q}_1, \boldsymbol{0})$ with

$$\boldsymbol{q}_0 = [1.6800, 1.2501, 2.4428, -1.2669, -0.9778, 1.1236, -1.3575]^\mathrm{T},$$
$$\boldsymbol{q}_1 = [2.7736, 0.5842, 1.5413, -1.7028, -2.1665, 0.0847, -2.5764]^\mathrm{T}.$$

See Figure 8 for an illustration of the task. Besides, we take the running cost and terminal cost to be

$$L(\boldsymbol{x}, \boldsymbol{u}) = \boldsymbol{a}(\boldsymbol{x}, \boldsymbol{u})^\mathrm{T} Q_{\boldsymbol{a}} \boldsymbol{a}(\boldsymbol{x}, \boldsymbol{u}) + (\boldsymbol{u} - \boldsymbol{u}_1)^\mathrm{T} Q_{\boldsymbol{u}} (\boldsymbol{u} - \boldsymbol{u}_1),$$
$$M(\boldsymbol{x}) = (\boldsymbol{x} - \boldsymbol{x}_1)^\mathrm{T} Q_f (\boldsymbol{x} - \boldsymbol{x}_1).$$

where $\boldsymbol{u}_1$ is the torque to balance gravity at state $\boldsymbol{x}_1$, *i.e.* $\boldsymbol{a}(\boldsymbol{x}_1, \boldsymbol{u}_1) = \boldsymbol{0}$. Under this setting, $(\boldsymbol{x}_1, \boldsymbol{u}_1)$ is an equilibrium of the system, *i.e.* $\boldsymbol{a}_1 = \boldsymbol{a}(\boldsymbol{x}_1, \boldsymbol{u}_1) = \boldsymbol{0}$ and $\boldsymbol{f}(\boldsymbol{x}_1, \boldsymbol{u}_1) = (\boldsymbol{v}_1, \boldsymbol{a}_1) = \boldsymbol{0}$.

In the experiment, we take $Q_{\boldsymbol{a}} = 0.005 I_7, Q_{\boldsymbol{u}} = 0.025 I_7, Q_f = 25000 I_{14}$ where we use large weights $Q_f$ to ensure the reaching goal is approximately achieved.

The backbone network for this example is the QRNet (Nakamura-Zimmerer et al., 2020; 2021b). QRNet exploits the solution corresponding to the LQR problem at equilibrium and thus improves the network performance around the equilibrium. The usage of other network structures also demonstrates the genericness/versatility of the IVP enhanced sampling method. Suppose we have the linear quadratic regulator (LQR) $\boldsymbol{u}^{\mathrm{LQR}}$ for the problem with linearized dynamics and quadratized costs at $(\boldsymbol{x}_1, \boldsymbol{u}_1)$, the QRNet can be formulated as

$$\boldsymbol{u}^{\mathrm{QR}}(t, \boldsymbol{x}) = \sigma(\boldsymbol{u}^{\mathrm{LQR}}(t, \boldsymbol{x}) + \boldsymbol{u}^{\mathrm{NN}}(t, \boldsymbol{x}; \theta) - \boldsymbol{u}^{\mathrm{NN}}(T, \boldsymbol{x}_1)),$$

where $\boldsymbol{u}^{\mathrm{NN}}(t, \boldsymbol{x}; \theta)$ is any neural network with trainable parameters $\theta$, and $\sigma$ is a saturating function that satisfies $\sigma(\boldsymbol{u}_1) = \boldsymbol{u}_1, \sigma_{\boldsymbol{u}}(\boldsymbol{u}_1) = I_7$. The $\sigma$ used in this example is defined coordinate-wisely as

$$\sigma(u) = u_{\min} + \frac{u_{\max} - u_{\min}}{1 + c_1 \exp[-c_2(u - u_1)]},$$

where $c_1 = (u_{\max} - u_1)/(u_1 - u_{\min}), c_2 = (u_{\max} - u_{\min})/[(u_{\max} - u_1)(u_1 - u_{\min})]$ with $u_{\min}, u_{\max}$ being minimum and maximum values for $u$. Here $u, u_{\min} = -150$ and $u_{\max} = 150$ are the corresponding values at each coordinate of $\boldsymbol{u}, \boldsymbol{u}_{\min}, \boldsymbol{u}_{\max}$, respectively.

To get the LQR, we need to expand the dynamics linearly as

$$\boldsymbol{f}(\boldsymbol{x}, \boldsymbol{u}) \approx \boldsymbol{f_x}(\boldsymbol{x}_1, \boldsymbol{u}_1)(\boldsymbol{x} - \boldsymbol{x}_1) + \boldsymbol{f_u}(\boldsymbol{x}_1, \boldsymbol{u}_1)(\boldsymbol{u} - \boldsymbol{u}_1),$$

and the term related to acceleration in the running cost quadratically as

$$\begin{aligned}
\boldsymbol{a}(\boldsymbol{x}, \boldsymbol{u})^{\mathrm{T}} Q_{\boldsymbol{a}} \boldsymbol{a}(\boldsymbol{x}, \boldsymbol{u}) &\approx \mathcal{L}_{\boldsymbol{a}}(\boldsymbol{x}, \boldsymbol{u})^{\mathrm{T}} Q_{\boldsymbol{a}} \mathcal{L}_{\boldsymbol{a}}(\boldsymbol{x}, \boldsymbol{u}) \\
&= (\boldsymbol{x} - \boldsymbol{x}_1)^{\mathrm{T}} \boldsymbol{a_x}^{\mathrm{T}} Q_{\boldsymbol{a}} \boldsymbol{a_x} (\boldsymbol{x} - \boldsymbol{x}_1) + (\boldsymbol{u} - \boldsymbol{u}_1)^{\mathrm{T}} \boldsymbol{a_u}^{\mathrm{T}} Q_{\boldsymbol{a}} \boldsymbol{a_u} (\boldsymbol{u} - \boldsymbol{u}_1) \\
&\quad + 2(\boldsymbol{x} - \boldsymbol{x}_1)^{\mathrm{T}} \boldsymbol{a_x}^{\mathrm{T}} Q_{\boldsymbol{a}} \boldsymbol{a_u} (\boldsymbol{u} - \boldsymbol{u}_1),
\end{aligned}$$

where $\mathcal{L}_{\boldsymbol{a}} = \boldsymbol{a_x}(\boldsymbol{x}_1, \boldsymbol{u}_1)(\boldsymbol{x} - \boldsymbol{x}_1) + \boldsymbol{a_u}(\boldsymbol{x}_1, \boldsymbol{u}_1)(\boldsymbol{u} - \boldsymbol{u}_1)$, and we exploit $\boldsymbol{a}(\boldsymbol{x}_1, \boldsymbol{u}_1) = \boldsymbol{0}$ and $\boldsymbol{f}(\boldsymbol{x}_1, \boldsymbol{u}_1) = \boldsymbol{0}$. The derivatives boil down to $\boldsymbol{a_x}$ and $\boldsymbol{a_u}$ which can be analytically computed in the Pinocchio library (Carpentier et al., 2015–2021; 2019; Carpentier & Mansard, 2018). In the experiment, we solve the LQR by the implementation in the Drake library (Tedrake & the Drake Development Team, 2019).

In the simulation and open-loop solver, we take time step $\Delta t = 0.001$ and use the semi-implicit Euler discretization. The initial positions $\boldsymbol{q}$ are sampled uniformly and independently in a 7-dimensional cube centered at $q_0$ with side length 0.02. Initial velocities $\boldsymbol{v}$ are set to zero. Other than directly applying the open-loop solver to collected initial states, we first sample another mini-batch of initial states and call the open-loop solver on it. We then pick one solution of the lowest cost and use it as an initial guess later. This can not only speed up the data generation process but also avoid sampling trajectories that fall into bad local minima. We use the differential dynamic programming solver implemented in (Mastalli et al., 2020), which is a second-order algorithm that favors a good initial guess. The dataset is then created from the (discrete) optimal trajectories warm-started by the initial guess. Each trajectory has $T/\Delta t = 800$ data points that are pairs of 15-dimensional input states including time and 7-dimensional output controls. The validation dataset and test dataset contain 605 and 1200 optimal trajectories, respectively.

Finally, all the QRNets $\boldsymbol{u}^{\mathrm{QR}}$ are trained by minimizing a mean square error (3) over the training dataset with the Adam optimizer (Kingma & Ba, 2015) with learning rate 0.001, batch size 256 and epochs 2000. $\boldsymbol{u}^{\mathrm{NN}}$ is a fully-connected network with 6 hidden layers; each layer has 128 neurons. The first three layers use the $\tanh$ function as activation while the last three layers use ELU (Clevert et al., 2016). Network and training are implemented in PyTorch (Paszke et al., 2019). During iterations, all networks are trained from scratch, *i.e.* a new network with random weights instead of inheriting weights from the previous iteration. After each epoch of training, we compute the loss on the validation dataset. The network with the least validation loss is then used for data generation in the next iteration or as the final policy (at the last iteration).

We compare the ratio of policy cost over the optimal cost, see Table 4 for the mean ratio over all 1200 trajectories of the test dataset. The ratio has been clipped at 2.0 for each trajectory. The results demonstrate that the IVP enhanced sampling method has great improvement over the vanilla supervised-learning-based method. It also shows the IVP enhanced sampling is not sensible to the choices of the temporal grid points. Besides, we also try augmenting the dataset with newly collected data instead of replacing them, as detailed in Section 8. Through the comparison between AS3 and AS3* in Table 4, we find that the alternative approach does not bring further improvement.

## G  COMPARISON WITH DAGGER

In this section, we give a comprehensive comparison between DAGGER (Dataset Aggregation) (Ross et al., 2011) and the IVP enhanced sampling, in terms of concepts, theoretical results, and numerical results. When referring to DAGGER, we mostly mean the single iteration version of DAGGER (*i.e.*, augmenting dataset once) unless otherwise stated explicitly.

**Concept.**  Both DAGGER and IVP enhanced sampling methods solve IVPs using the policy from the previous iteration to generate time-state pairs as new initial time-state pairs and call the open-loop solver to label the trajectories starting from these pairs. Their main differences are as follows.

| | # of iterations | Temporal grid points ($T = 0.8$) | | | | | |
|---|---|---|---|---|---|---|---|
| | | 0.0 | 0.16 | 0.48 | 0.56 | 0.64 | 0.72 |
| **Vanilla300** | 1 | 1.9204 | | | | | |
| **Vanilla900** | 1 | 1.8717 | | | | | |
| **AS1** | 3 | 1.8104 | 1.2830 | 1.1182 | | | |
| **AS2** | 3 | 1.9971 | 1.4535 | | 1.0441 | | |
| **AS3** | 3 | 1.9161 | 1.3678 | | | 1.0745 | |
| **AS3\*** | 3 | 1.8549 | 1.4402 | | | 1.1491 | |
| **AS4** | 3 | 1.9552 | 1.2892 | | | | 1.1295 |

Table 4: The mean ratio of policy costs / optimal costs of the optimal reaching problem of the manipulator. The ratio has been clipped at 2.0 for each test trajectory. The vanilla300/900 correspond to networks trained on 300/900 optimal trajectories, respectively. The choices of temporal grid points for adaptive sampling in the remaining rows can be inferred by the location of columns. For example, AS1 has temporal grid points $0 < 0.16 < 0.48 < 0.8$. AS3* has the same temporal grid points as AS3 except that it augments the dataset directly instead of replacing them, as discussed in Section 8.

In the $i$-th iteration, the IVP enhanced sampling method only solves the IVPs till the $i$-th time grid and uses the time and the visited states at the $i$-th time grid as the initial time-state pairs to collect new open-loop optimal data. In contrast, in each iteration, the DAGGER method needs to solve until the penultimate temporal grid $t_{K-1}$, collect the states on all the time grids $0 < t_1 < \cdots < t_{K-1} < T$, and use all the collected time-state pairs as the initial time-state pair to generate new open-loop optimal data. In the IVP enhanced sampling method, the later times are only visited by networks trained from later iterations. Hopefully, networks from later iterations perform better and generate relevant time-state pairs at later time grids. In contrast, each network from the DAGGER method acts on the dynamical system until the final grid. For stiff dynamics that accumulate errors fast, the time-state pairs at later time grids generated by a network from earlier iterations can deviate much from what the optimal policy will visit. These data might then deteriorate the performance instead, as supported by the numerical results below.

**Computation cost.** With the same time grids (say $0 = t_0 < t_1 \cdots < t_K = T$), we argue that the efforts of training the IVP enhanced sampling method and single iteration DAGGER are approximately the same since the cost for data labeling and training are approximately the same, respectively.

First, in terms of the computation cost of data labeling, the IVP enhanced sampling method requires labeling $K$ dataset of $M$ trajectories. The single iteration DAGGER labels the same amounts of data. Though DAGGER requires fewer efforts in solving the IVPs as the IVP enhanced sampling method always solves IVPs from $t = 0$. However, for many problems, the time spent in solving IVPs is negligible compared to that in solving the open-loop optimal control problems. Specifically, in our first numerical example of optimal landing, it takes about 2.5 hours to do the BVP computation in total while it only takes 7 minutes to do IVP integration in total. In the second numerical example of the reaching problem, to solve 100 trajectories, it takes 308 seconds to solve the open-loop solution through DDP while it only takes 3 seconds to do IVP integration. Therefore, the cost for data labeling is approximately the same.

Second, in terms of training time, let us assume that the initial dataset contains $M$ trajectories and each trajectory contributes $N$ time-state-action tuples. The IVP enhanced sampling method needs to train $K$ networks; each network is trained on an enhanced dataset with $M$ trajectories. Then, there are in total $MKN$ time-state-action tuples visited in the IVP enhanced sampling method (same data is counted repeatedly when training different networks, the same below). DAGGER trains two networks, one with $M$ trajectories and the other with $MK$ trajectories. The former dataset contains $MN$ time-state-action tuples while the latter one contains approximately

$$M(\frac{T - t_0}{T} + \frac{T - t_1}{T} + \cdots + \frac{T - t_{K-1}}{T})N = M(K - \frac{1}{T}(t_0 + t_1 + \cdots + t_{K-1}))N$$

time-state-action tuples. Then, for time grids $t_0 = 0 < 10 < 14 < 16 = T$ (*e.g.* the landing problem of a quadrotor below), DAGGER visits approximately 16.67% fewer data than the IVP

enhanced sampling method; for time grids $t_0 = 0 < 0.16 < 0.64 < 0.8 = T$ and $t_0 = 0 < 0.16 < 0.48 < 0.8 = T$ (*e.g.* the reaching problem of the manipulator below), DAGGER visits the same amounts of data as and approximately $6.67\%$ more than the IVP enhanced sampling method, respectively; Therefore, with the same number of epochs in training each network, the training efforts do not differ much in the two methods.

In the following, we compare the IVP enhanced sampling method with DAGGER. We will see that, for the LQR example in Section 5 and Appendix B, the IVP enhanced sampling method surpasses DAGGER both theoretically and numerically, especially for large $T$. For the landing and reaching problem studied previously (see Appendix F and E), both methods perform similarly well. However, in more difficult settings of both problems, the IVP enhanced sampling method outperforms DAGGER.

**Results on the LQR example.** We first investigate the performance of DAGGER on the LQR example. With a slight abuse of notation, we will use $\hat{x}_j^i(t)$ and $\hat{u}_j^i(t)$ to denote the open-loop optimal paths sampled for training, $\tilde{x}_j^i$ to denote the initial states used to generate the training trajectories, $u_d(t, x)$ to denote the closed-loop controller learned by DAGGER, and $x_d(t)$, $u_d(t)$ to denote the IVP solution generated by the closed-loop controller $u_d$.

In DAGGER, we again choose $K = T$ and the temporal grid points $t_i = i$ for $0 \le i \le K$. We first sample $N$ initial points $\{\tilde{x}_j^0\}_{j=1}^N$ from the normal standard distribution and then generated $N$ approximated optimal paths starting at $t_0 = 0$:

$$\hat{u}_j^0(t) = -\frac{T}{T^2 + 1}\tilde{x}_j^0 + \epsilon Z_j^0, \hat{x}_j^0(t) = \frac{T(T - t) + 1}{T^2 + 1}\tilde{x}_j + \epsilon t Z_j^0,$$

where $\{Z_j^0\}_{j=1}^N$ are *i.i.d.* normal random variables and independent with initial states whose mean is $m$ and variance is $\sigma^2$. We then train the closed-loop controller $u_0$ by solving the following least square problems:

$$\min_\theta \int_0^T \sum_{j=1}^N |\hat{u}_j^0(t) - u_0(t, \hat{x}_j^0(t))|^2 \mathrm{d}t.$$

Then, we use $u_0$ to solve the IVPs on the whole time horizon $[0, T]$ with initial states $\{\tilde{x}_j^0\}$:

$$\dot{x}_j^0(t) = u_0(t, x_j^0(t)), x_j^0(0) = \tilde{x}_j^0, 1 \le j \le N, \tag{25}$$

and collect $\{\tilde{x}_j^i\}_{j=1}^N$ as $\tilde{x}_j^i := x_j^0(i)$ for $i = 1, 2, \ldots, T - 1$. At each time step $t_i = i$, we then compute $N$ approximated optimal paths starting from $\{\tilde{x}_j^i\}_{j=1}^N$:

$$\hat{u}_j^i(t) = -\frac{T}{T(T - i) + 1}\tilde{x}_j^i + \epsilon Z_j^i, \hat{x}_j^i(t) = \frac{T(T - t) + 1}{T(T - i) + 1}\tilde{x}_j^i + (t - i)\epsilon Z_j^i, t \in [i, T] \tag{26}$$

where $\{Z_j^i\}_{1 \le i \le T-1}$ are *i.i.d.* normal random variables and independent with $\{\tilde{x}_j^0\}_{j=1}^N$ and $\{Z_j\}_{j=1}^N$ whose mean is $m$ and variance is $\sigma^2$. Finally, we collect the optimal paths $\{(\hat{u}_j^i, \hat{x}_j^i)\}_{0 \le i \le T-1, 1 \le j \le N}$ to train the closed-loop controller $u_d$ by solving the following least square problems:

$$\min_\theta \int_i^{i+1} \sum_{k=0}^i \sum_{j=1}^N |\hat{u}_j^k(t) - u_\theta(t, \hat{x}_j^k(t))|^2 \mathrm{d}t \tag{27}$$

for $k = 0, 1, \ldots, T - 1$.

**Theorem 2.** *Under Model 1* (4), *define IVP generated by $u_d$:*

$$\dot{x}_d(t) = u_d(t) = u_d(t, x_d(t)).x_d(t) = \tilde{x}_{init}, 0 \le t \le T \tag{28}$$

*and the total cost:*

$$J_d = \frac{1}{T}\int_0^T |u_d(t)|^2 \mathrm{d}t + |x_d(T)|^2.$$

*If $\tilde{x}_{init}$ is a fixed point, then*

$$\mathbb{E}J_d - J_o \ge (\frac{T^2 m^2}{4} + \frac{T\sigma^2}{3N})\epsilon^2.$$

*Proof.* With the same approach of computing $u_v$ in (16), we have

$$u_0(t, x) = -\frac{T}{T(T - t) + 1} x + \frac{T^2 + 1}{T(T - t) + 1} \epsilon \bar{Z}_d^0,$$

where

$$\bar{Z}_d^0 = \frac{1}{N} \sum_{j=1}^{N} Z_j^0.$$

Recalling the definition of $x_j^0$ in equation (25), we have

$$x_j^0(t) = \frac{T(T - t) + 1}{T^2 + 1} \tilde{x}_j^0 + \epsilon t \bar{Z}_d^0.$$

Hence, for $0 \le i \le T - 1$, we have

$$\tilde{x}_j^i = x_j^0(i) = \frac{T(T - i) + 1}{T^2 + 1} \tilde{x}_j^0 + \epsilon i \bar{Z}_d^0.$$

Plugging the last equation into equation (26), we have that for $t \in [i, T]$

$$\hat{u}_j^i(t) = -\frac{T}{T^2 + 1} \tilde{x}_j^0 - \epsilon \frac{Ti}{T(T - i) + 1} \bar{Z}_d^0 + \epsilon Z_j^i,$$

$$\hat{x}_j^i(t) = \frac{T(T - t) + 1}{T^2 + 1} \tilde{x}_j^0 + \epsilon \frac{iT(T - t) + i}{T(T - i) + 1} \bar{Z}_d^0 + (t - i) \epsilon Z_j^i.$$

Therefore,

$$\hat{u}_j^i(t) = -\frac{T}{T(T - t) + 1} \hat{x}_j^i(t) + \frac{T(T - i) + 1}{T(T - t) + 1} \epsilon Z_j^i.$$

We can then compute the least square problem (27) to obtain that for $0 \le i \le T - 1$ and $t \in [i, i+1)$, we have

$$u_d(t, x) = -\frac{T}{T(T - t) + 1} x + \frac{1}{i + 1} \sum_{k=0}^{i} \frac{T(T - k) + 1}{T(T - t) + 1} \epsilon \bar{Z}_d^k,$$

where

$$\bar{Z}_d^i = \frac{1}{N} \sum_{j=1}^{N} Z_j^i,$$

for $0 \le i \le T - 1$. We can then solve the ODE (28), we have that when $0 \le i \le T - 1$ and $t \in [i, i+1)$

$$x_d(t) = \frac{T(T - t) + 1}{T^2 + 1} \tilde{x}_{\text{init}} + \sum_{k=0}^{i-1} \frac{F(t)}{F(k+1)F(k)} \frac{1}{k+1} \sum_{l=0}^{k} \epsilon F(l) \bar{Z}_d^l + \frac{t - i}{(i + 1)F(i)} \sum_{k=0}^{i} F(k) \epsilon \bar{Z}_d^k,$$

where $F(t) = T(T - t) + 1$. Therefore, when $0 \le i \le T - 1$ and $t \in [i, i+1)$,

$$u_d(t) = -\frac{T}{T^2 + 1} \tilde{x}_{\text{init}} - \sum_{k=0}^{i-1} \frac{T}{(k+1)F(k)F(k+1)} \sum_{l=0}^{k} \epsilon F(l) \bar{Z}_d^l + \frac{1}{(i + 1)F(i)} \sum_{k=0}^{i} F(k) \epsilon \bar{Z}_d^k.$$

Define

$$e_i = -\sum_{k=0}^{i-1} \frac{T}{(k+1)F(k)F(k+1)} \sum_{l=0}^{k} \epsilon F(l) \bar{Z}_d^l + \frac{1}{(i + 1)F(i)} \sum_{k=0}^{i} F(k) \epsilon \bar{Z}_d^k, 0 \le i \le T - 1$$

then

$$J_d = \frac{1}{T} \sum_{i=0}^{T-1} \left| -\frac{T}{T^2 + 1} \tilde{x}_{\text{init}} + e_i \right|^2 + \left| \tilde{x}_{\text{init}} - \frac{T^2}{T^2 + 1} \tilde{x}_{\text{init}} + \sum_{i=0}^{T-1} e_i \right|^2$$

$$= \frac{T^2 |x_{\text{init}}|^2}{(T^2 + 1)^2} - \sum_{i=0}^{T-1} \frac{2 \tilde{x}_{\text{init}} e_i}{T^2 + 1} + \frac{1}{T} \sum_{i=0}^{T-1} |e_i|^2 + \frac{|\tilde{x}_{\text{init}}|^2}{(T^2 + 1)^2} + \sum_{i=0}^{T-1} \frac{2 e_i \tilde{x}_{\text{init}}}{T^2 + 1} + \left| \sum_{i=0}^{T-1} e_i \right|^2$$

$$= \frac{|\tilde{x}_{\text{init}}|^2}{T^2 + 1} + \frac{1}{T} \sum_{i=0}^{T-1} |e_i|^2 + \left| \sum_{i=0}^{T-1} e_i \right|^2.$$

Therefore

$$\mathbb{E}J_d - J_o \geq \mathbb{E}|\sum_{i=0}^{T-1} e_i|^2 = (\mathbb{E}\sum_{i=0}^{T-1} e_i)^2 + \text{Var}(\sum_{i=0}^{T-1} e_i). \tag{29}$$

We can then compute that

$$\sum_{i=0}^{T-1} e_i = -\sum_{i=0}^{T-1}\sum_{k=0}^{i-1}\sum_{l=0}^{k} \frac{\epsilon T F(l)\bar{Z}_d^l}{(k+1)F(k)F(k+1)} + \sum_{i=0}^{T-1}\sum_{k=0}^{i} \frac{F(k)\epsilon\bar{Z}_d^k}{(i+1)F(i)}$$

$$= \sum_{i=0}^{T-1}\sum_{k=0}^{i} \frac{F(k)\epsilon\bar{Z}_d^k}{(i+1)F(i)} - \sum_{k=0}^{T-1}\sum_{l=0}^{k}\sum_{i=k+1}^{T-1} \frac{\epsilon T F(l)\bar{Z}_d^l}{(k+1)F(k)F(k+1)}$$

$$= \sum_{i=0}^{T-1}\sum_{k=0}^{i} \frac{F(k)\epsilon\bar{Z}_d^k}{(i+1)F(i)} - \sum_{i=0}^{T-1}\sum_{k=0}^{i} \frac{\epsilon T(T-i-1)F(k)\bar{Z}_d^k}{(i+1)F(i)F(i+1)}$$

$$= \sum_{i=0}^{T-1}\sum_{k=0}^{i} \frac{F(k)\epsilon\bar{Z}_d^k}{(i+1)F(i)F(i+1)}.$$

Therefore,

$$\mathbb{E}\sum_{i=0}^{T-1} e_i = \epsilon m \sum_{i=0}^{T-1}\sum_{k=0}^{i} \frac{F(k)}{(i+1)F(i)F(i+1)} \tag{30}$$

$$= \epsilon m \sum_{i=0}^{T-1} \frac{(T^2+1)(i+1) - Ti(i+1)/2}{(i+1)F(i)F(i+1)}$$

$$= \epsilon m \sum_{i=0}^{T-1} \frac{T^2+1-Ti/2}{[T(T-i)+1][T(T-i-1)+1]}$$

$$\geq \epsilon m \frac{T^2+T+2}{2T} \sum_{i=0}^{T-1}[\frac{1}{T(T-i-1)+1} - \frac{1}{T(T-i)+1}]$$

$$= \epsilon m \frac{T^2+T+2}{2T}(1 - \frac{1}{T^2+1}) \geq \frac{\epsilon m T}{2}.$$

On the other hand, noticing that

$$\sum_{i=0}^{T-1} e_i = \sum_{k=0}^{T-1}\sum_{i=k}^{T-1} \frac{F(k)\epsilon\bar{Z}_d^k}{(i+1)F(i)F(i+1)},$$

we have

$$\text{Var}(\sum_{i=0}^{T-1} e_i) = \frac{\epsilon^2\sigma^2}{N}\sum_{k=0}^{T-1} F^2(k)(\sum_{i=k}^{T-1} \frac{1}{(i+1)F(i)F(i+1)})^2 \tag{31}$$

$$\geq \frac{\epsilon^2\sigma^2}{NT^4}\sum_{k=0}^{T-1}[T(T-k)+1]^2(\sum_{i=k}^{T-1} \frac{1}{T(T-i-1)+1} - \frac{1}{T(T-i)+1})^2$$

$$= \frac{\epsilon^2\sigma^2}{NT^4}\sum_{k=0}^{T-1}[T(T-k)+1]^2[1 - \frac{1}{T(T-k)+1}]^2$$

$$= \frac{\epsilon^2\sigma^2}{NT^2}\sum_{k=0}^{T-1}(T-K)^2 = \frac{\epsilon^2\sigma^2}{NT^2}\frac{T(T+1)(2T+1)}{6} \geq \frac{\epsilon^2\sigma^2 T}{3N}.$$

Combining equations (29), (30) and (31), we can conclude our result. □

We then numerically compare the performance of DAGGER with the vanilla method and the IVP enhanced sampling method on Model 2 (5). In these experiments, we again set $\epsilon = 0.1$, $m = 0.1$

and $\sigma^2 = 1$. We present the paths generated by the optimal controller and the controllers learned by the vanilla method, the IVP enhanced sampling method, and DAGGER in Figure 9 (left). Figure 9 (middle) compares the performance of these three methods on different time horizons $T$. We also test DAGGER with multiple iterations in Figure 9 (right). Here we set total time $T = 30$. The experiment shows that the performance of the learned controller does not improve with more iterations. The detailed settings are identical to the numerical experiments in Appendix B.

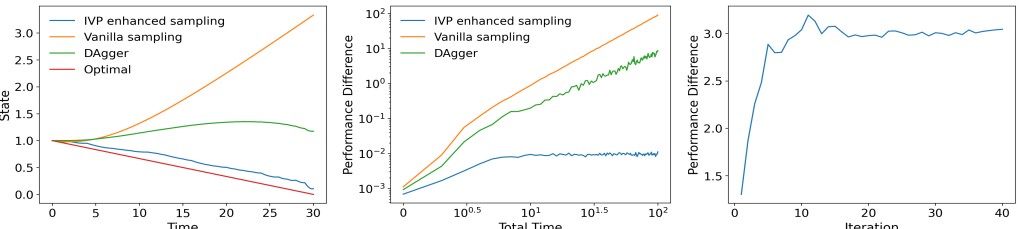

Figure 9: Numerical results on learning Model 2 (5) with DAGGER.

**Results on the landing problem.**   Following the same settings as in Appendix E, we train a policy by DAGGER with additional sampling at time $t = 10$ and $14$. The results are summarized in Table 5. It shows that DAGGER performs closely to the IVP enhanced sampling method. However, if we decrease the number of trajectories in the initial dataset from $500$ to $300$ to train controllers using these two methods, we observe more decreases in performance in the DAGGER method, which implies that DAGGER is more sensitive to the data amount. The main reason is that DAGGER demands enough data at the beginning to have a good initial controller to explore the state over the whole time interval. However, in complicated control problems, we do not have such privilege and indeed require adaptive sampling to improve the controller.

|  | # **of iterations** | result |
|---|---|---|
| **AS** | 3 | 1.092 |
| **DAGGER** | 2 | 1.069 |
| **AS-300** | 3 | 1.287 |
| **DAGGER-300** | 2 | 1.405 |

Table 5: Average cost ratio on 200 test points of controllers trained by the IVP enhanced sampling method and DAGGER. All models are with time grids $0 < 10 < 14 < 16$. Models with the suffix 300 are those trained on the initial dataset with 300 trajectories, whose cost ratios are clipped at 10.0 for each test trajectory.

**Results on the reaching problem.**   For the problem detailed in Appendix F, by additionally sampling at time $0.16$ and $0.64$ in one iteration, the DAGGER algorithm achieves a policy cost / optimal cost ratio of $1.049$ on the test dataset, which is close to that achieved by the IVP enhanced sampling method.

We then increase the difficulty of the control problem by increasing the moving distance. Following the configurations in Appendix F, we change the center of initial position and the terminal position to

$$q_0 = [1.60, 1.30, 2.70, -0.85, -1.90, 0.95, -1.60]^\mathrm{T},$$

$$q_1 = [2.75, 0.60, 2.00, -1.55, -2.15, 0.00, -2.60]^\mathrm{T}.$$

Besides, as the DAGGER will generate states in a wider range, we modify $u_{\min}, u_{\max}$ in QRNet to $u_{\min} = -2000, u_{\max} = 2000$ in order to avoid saturation. We also increase the size of the initial dataset from 100 trajectories to 200 trajectories. Each network is trained with 1500 epochs. The other settings are the same as that in Appendix F.

Each method has been run 5 times independently, and we report their average and best performance. The results are summarized in Table 6 and Figure 10. As we can see, the IVP enhanced sampling

method is capable of finding a closed-loop controller with an average ratio between policy cost and optimal cost achieving 1.0155. However, the DAGGER algorithm cannot yield such a satisfactory result.

In DAGGER1, we apply the DAGGER method with time grids $0 < 0.16 < 0.48 < 0.8$, an earlier final grid compared to DAGGER2 which has time grids $0 < 0.16 < 0.64 < 0.8$. We see an average improvement from 1.8528 to 1.6327. DAGGER1 performs similarly to the network trained at the second iteration of AS, which implies that the extra data sampled at $t = 0.48$ does not help much. Furthermore, we conduct an additional iteration of the DAGGER method, which requires solving the open-loop problem to get 400 more trajectories and extra training of the network with 1000 trajectories in total. It performs worse, which also confirms the arguments we made at the beginning that the uncarefully collected data may deteriorate the performance.

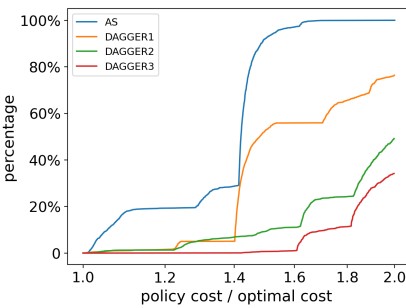 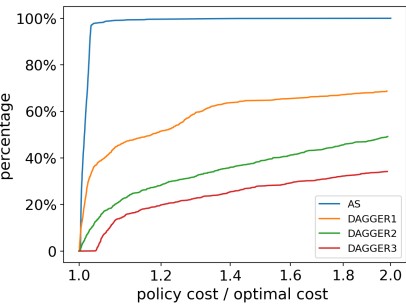

Figure 10: Cumulative distribution functions of average cost ratios over 5 independent experiments (left) and cost ratios of the best controller among 5 independent experiments under the proposed method and DAGGER for the optimal reaching problem with a larger moving distance.

|  | # **of iterations** | **Temporal grid points** ($T = 0.8$) | | |
|---|---|---|---|---|
|  |  | 0.16 | 0.48 | 0.64 |
| **AS** | 3 | 1.6592 (1.5417) | | 1.3610 (1.0155) |
| **DAGGER1** | 2 | | 1.6327 (1.4004) | |
| **DAGGER2** | 2 | | | 1.8528 (1.6364) |
| **DAGGER3** | 3 | | | 1.9293 (1.7478) |

Table 6: The mean ratio between policy costs and optimal costs of the reaching problem with larger moving distance. In each cell, the first number is averaged over 5 independent experiments and the number in parenthesis is the average ratio achieved by the best controller among 5 independent experiments. The ratio has been clipped at 2.0 for each test trajectory. Both AS and DAGGER2 have temporal grid points $0 < 0.16 < 0.64 < 0.8$. DAGGER3 repeats DAGGER2 for one more DAGGER iteration.

