# OpenReview forum: "Initial Value Problem Enhanced Sampling for Closed-Loop Optimal Control Design with Deep Neural Networks"
_ICLR.cc/2023/Conference — Submitted to ICLR 2023_

### Official Review · Reviewer_gCS8 · 2022-10-20

**Confidence:** 4
**Correctness:** 3
**Technical Novelty And Significance:** 2
**Empirical Novelty And Significance:** 2
**Recommendation:** 6

**Clarity, Quality, Novelty And Reproducibility:**

Clarity
---------
The presentation of the approach is clear.  However, I am missing some details in the experimental section:
How is the neural network controller optimized? In particular: how long was it trained? Was a validation set used for early stopping? If not, how was it ensured, that the improved performance was due to the change in the training set, and not simply due to longer training?
Furthermore, I was surprised that no state-trajectories where shown for the quadrotor experiment. I think it would be important to show that the data generated by the early controllers is still relevant at later iterations.

Quality
---------
- The theoretical analysis is rather weak, as it only applies for a simplified LQR  assuming that the state error of the closed-loop controller is zero mean. While I see the merits of using LQR for tractable analysis, the assumption of zero mean error seems too strong and unrealistic.
- It does not seem principled to throw away optimal labels that were quite costly to obtain. Why are parts of the training data replaced by the newly collected data, instead of augmenting the data set? The paper does not discuss this option at all, nor does it show experimental comparisons. In general, the paper does not provide motivation for the *specific* way the additional data is considered, and what are the benefits compared to alternate methods.

Novelty
---------
- The overall procedure of iteratively selecting initial states for trajectory optimization in order to obtain data to improve the closed-loop controller is not novel (the paper already contains several references). So the only algorithmic contribution relates to the specific way these points are chosen.  While the paper discusses the forward training algorithm, it does not discuss its follow up DAGGER (Ross et al., 2011) which might be even more related. The analogous method of DAGGER in continuous time, would proceed as follow.
1. solve optimal control from the initial states, and train close-loop controller (same as the proposed method)
2. solve IVP to obtain the states at each temporal grid point $t_i$
3. solve optimal control for each of these states and augment dataset, retrain close-loop controller
4. Goto 2, if necessary

I would argue, that such approach would not be that different from the proposed method. The difference would be that for every solved IVP, it would need to solve $K$ trajectory optimization problems (one per grid point). Performing a single iteration (i.e. skipping step 4) would require as many calls to the trajectory optimizer as the proposed method, but fewer calls to the IVP-solver. If the IVP can not be solved analytically, but requires numerical integration, I could imagine that the DAGGER approach can be more cost-effective.

Reproducibility
--------------------
Without source code it is not possible to reproduce the results, as details---in particular related to training the NN controller---are missing in the paper.


References
---------------
Ross, Stéphane, Geoffrey Gordon, and Drew Bagnell. "A reduction of imitation learning and structured prediction to no-regret online learning." Proceedings of the fourteenth international conference on artificial intelligence and statistics. JMLR Workshop and Conference Proceedings, 2011.

**Strength And Weaknesses:**

Strength
----------
- The paper is well-written and clear. The presentation is good.
- The claims seem correct.

Weaknesses
-----------------
- The novelty is rather limited (details below).
- The theoretical results are weak (details below).

**Summary Of The Paper:**

The paper considers the problem of learning (neural network) closed-loop controllers using supervised learning based on data that is generated by trajectory optimization.
To mitigate the well-known problem of distribution mismatch, the paper proposes an iterative procedure. The paper assumes that a few time steps $t_i$ are specified to split the trajectory into different segments. In the first iteration, trajectory optimization is used to generate trajectories starting at time step $t_0=0$, which form the training data for generate the neural network controller. In the second iteration the training data from [t_0, t_1] are kept, but the data from [t1, T] are replaced by optimal trajectories that start at t_1 from states reached by the NN-controller (which can be found by approximately solving the corresponding initial value problem). This process is repeated until each of the pre-specified time-steps had served as initial time-step.

This procedure of generating training data for the closed-loop controller seems novel, although similar methods have been proposed in the past. Furthermore, the paper provides a theoretical analysis on a specific LQR problem showing that the iterative data collection leads to smaller state-error. The method is evaluated for learning controllers for landing a quadrotor and for performing a reaching motions for a robotic arm (both in simulation).

**Summary Of The Review:**

The paper is clear and seems technically correct. However, the contributions seem rather small, as the provided theoretical, empirical and intuitive justifications are rather weak. The paper could be improved by evaluating the effect of keeping the data of previous iterations, and by evaluating the DAGGER-style procedure (comparisons should be over computational cost/time).

---

> ### Author Response · Authors · 2022-11-18
> **Reply to reviewer gCS8**
>
> We thank you a lot for your review. We have uploaded a new version of the paper with changes based on your feedback. All essential changes are highlighted in blue color. We believe that the revision has significantly improved the quality of our paper. We hope those revisions resolve your concerns, thus improving your evaluation of our work and may make you consider raising the score. Below, we reply to your concerns point-by-point and summarize the related changes.
>
> > **Q1.** The theoretical analysis is rather weak, as it only applies for a simplified LQR assuming that the state error of the closed-loop controller is zero mean. While I see the merits of using LQR for tractable analysis, the assumption of zero mean error seems too strong and unrealistic.
>
> In the revision, we have generalized our theoretical analysis by considering the state error comes from a general normal random variable with a non-zero mean. The new results are restated in Theorem 1 in the main text and Theorem 1' in Appendix B. We can clearly see that the advantages of the IVP enhanced sampling over the vanilla sampling method remain the same in this more general setting.
>
>
>
> > **Q2.** The overall procedure of iteratively selecting initial states for trajectory optimization in order to obtain data to improve the closed-loop controller is not novel (the paper already contains several references). So the only algorithmic contribution relates to the specific way these points are chosen. While the paper discusses the forward training algorithm, it does not discuss its follow up DAGGER (Ross et al., 2011) which might be even more related. I would argue, that such approach would not be that different from the proposed method. The difference would be that for every solved IVP, it would need to solve  trajectory optimization problems (one per grid point). Performing a single iteration (i.e. skipping step 4) would require as many calls to the trajectory optimizer as the proposed method, but fewer calls to the IVP-solver. If the IVP can not be solved analytically, but requires numerical integration, I could imagine that the DAGGER approach can be more cost-effective.
>
> Thanks to the reviewer for pointing out this relevant literature. In the revision, we have added a paragraph in Section 4 discussing the DAGGER method and a new appendix section to compare DAGGER with the IVP enhanced sampling theoretically and numerically. In summary, the IVP enhanced sampling method performs better.
>
>
> In DAGGER, in order to improve the current closed-loop controller $\hat{\boldsymbol{u}}$, one solves IVPs using $\hat{\boldsymbol{u}}$ over $[0, T]$ starting from various initial states and collect the states on a time grid $0 < t_1 < \dots < t_{K-1} < T$. The open-loop optimal control problems are then solved with all the collected time-state pairs as the initial time-state pairs, and all the corresponding optimal solutions are used to construct a dataset for learning a new controller. The process can be repeated until a good controller is obtained. The time-state selection in DAGGER is also related to the distribution mismatch phenomenon, but somehow different from the IVP enhanced sampling. Take the data collection using the controller $\hat{\boldsymbol{u}}_1$ in the first iterative step for an example. The IVP enhanced sampling focuses on the states at the time grid $t_1$ while DAGGER collects states at all the time grids. If $\hat{\boldsymbol{u}}_1$ is still far from optimal, the data collected at later time grids may be irrelevant to or even mislead training due to error accumulation in states. In Appendix G, we first theoretically analyze the DAGGER method in our LQR setting and find that its performance gap has the same scaling as the vanilla method in terms of total horizon $T$, which is much worse than the IVP enhanced sampling method. Furthermore, the numerical results in both quadrotor and manipulator problems show that our method gives better results.
>
> Regarding your concern about the computation cost of IVP solving, we would like to point out that the total computation time related to IVP takes a small portion compared to the open-loop optimal solver. Specifically, in our first numerical example, it takes about 2.5 hours to do the BVP computation in total while it only takes 7 minutes to do IVP integration in total. In our second numerical example, to solve 100 trajectories, it takes 308 seconds to solve the open-loop solution through DDP while it only takes 3 seconds to do IVP integration.

---

> > ### Author Response · Authors · 2022-11-18
> > **Reply to reviewer gCS8**
> >
> > **(continued)**
> > > **Q3.** It does not seem principled to throw away optimal labels that were quite costly to obtain. Why are parts of the training data replaced by the newly collected data, instead of augmenting the data set? The paper does not discuss this option at all, nor does it show experimental comparisons. In general, the paper does not provide motivation for the specific way the additional data is considered, and what are the benefits compared to alternate methods.
> >
> > We do admit that the way we use newly collected data is not the only choice. In the last section of the revised version, we have added a discussion on this issue. Specifically, In our Algorithm 1 (lines 8--9), at each iteration, we replace parts of the training data with the newly collected data and hence some optimal labels are thrown away. An alternative choice is to augment data directly, i.e., setting $S_i={\hat{{S}}_{i}} \cup S\_{i-1}$ in line 9. Numerically, we observe that this choice gives a similar performance to the version used in Algorithm 1 (see Table 1 in Appendix E and Table 4 in Appendix F), which suggests that so far the dropped data provides little value for training. But it is still possible to find smarter ways to utilize them to improve the performance.
> >
> >
> > > **Q4.** I am missing some details in the experimental section: How is the neural network controller optimized? In particular: how long was it trained? Was a validation set used for early stopping? If not, how was it ensured, that the improved performance was due to the change in the training set, and not simply due to longer training? Without source code it is not possible to reproduce the results, as details---in particular related to training the NN controller---are missing in the paper.
> >
> > We have added the missing details in Appendix E, F, and G. All neural networks are independently trained from scratch for a fixed number of epochs. They are randomly initialized and do not inherit weights from the previous network. In the quadrotor experiment, no validation set is used and we train all networks with 1000 epochs. In the manipulator experiment, the problem is more difficult, and thus the IVP enhanced sampling method is more sensitive to the quality of the networks during the iterations. We introduce a validation dataset to help select the network. After each training epoch, we compute the loss on the validation dataset. The network with the least validation loss is then selected.
> >
> > > **Q5.** Furthermore, I was surprised that no state-trajectories where shown for the quadrotor experiment. I think it would be important to show that the data generated by the early controllers is still relevant at later iterations.
> >
> > In the revision, we have added a new figure, Fig 6 in Appendix E, to illustrate the trajectories of states controlled by learned policy and optimal policy. The path controlled by the policy after iteration 0 matches the optimal path at the beginning but deviates around t = 10. Then the path controlled by the policy after iteration 1 fits the optimal path more and deviates around t = 14.
> > Finally, the path controlled by the policy after iteration 2 matches the optimal path almost till the terminal time. Compared to the vanilla/initial/first training dataset, the final training data set explores more states but does not deviate too much. Note that the cost of three controlled paths is 3296.18, 119.91, 6.69, respectively, and the optimal cost is 6.32.

---

### Official Review · Reviewer_s6p1 · 2022-10-23

**Confidence:** 2
**Correctness:** 4
**Technical Novelty And Significance:** 3
**Empirical Novelty And Significance:** 3
**Recommendation:** 6

**Clarity, Quality, Novelty And Reproducibility:**

As mentioned above, I have some concerns about reproducibility given the reliance on determining the sampling intervals. However, the others have acknowledged this in the conclusion so I think it's reasonable to not have a clearer process for this yet. Beyond this I thought the writing was clear and of high quality.

In terms of novelty, I am not too familiar with the related work, so I can't evaluate this too well. The problem of distribution mismatch seems somewhat obvious to me since it plagues many ML/RL-related problems, but as far as I can tell this is a novel solution to that issue.

**Strength And Weaknesses:**

Strengths:
- Points out a weakness in previous works and proposes a novel solution to it.
- Writing is intuitive, and a good mix of theoretical and empirical results

Weaknesses:
- Main concern is the reliance on determining the sampling intervals when training a model. This seems like a critical hyperparameter and I don't have a sense for how easy it is to tune, and hence I'm not sure how easy it would be to extend the approach to other problem domains.

**Summary Of The Paper:**

This paper presents a new method for learning deep closed-loop control policies from training on data generated by open-loop optimal control solvers. The paper proposes a resampling procedure that iteratively retrains the controller in states that it may not have seen in the initial training data. The approach was tested empirically in two problem domains and was shown to achieve good performance and to be somewhat robust to dynamics noise.

**Summary Of The Review:**

This paper identifies a problem in an existing line of work, proposes a solution to that problem, and demonstrates the value of that solution theoretically and empirically. Taken together, I think that makes this a worthwhile read and worthy of acceptance, although I will admit that this specific area of learning closed-loop controllers from open-loop data is new to me, so I would defer to others with more knowledge of the related work for their judgments.

Some comments/questions:
- The problem setup here involves applying control to a system over some finite interval [0, T]. How would this approach extend to a system that needs to operate continuously (e.g. a robot in a factory) if the premise here is that eventually control quality begins to degrade? Would you need to break up tasks into sub-tasks?
- The problem formulation here assumes that we have knowledge of the system dynamics. What if those models are inaccurate? I would think that you would end up with a similar problem to the one you used to motivate this work, i.e. that the controls you apply bring you to states you haven't seen in your training data and that degrades performance more and more over time. Given that, do you think this approach could be extended to problems where dynamics models are inaccurate? I feel like this theme was somewhat explored in the experiments where noise was added to the dynamics, but it wasn't explicitly framed in this way
- In the iterative training procedure do you use the same network throughout and finetune that network during each training stage or do you train a new network during each stage. It may have said this somewhere in the paper and I missed it, but it might be nice to state this more clearly e.g. within Algorithm 1.

---

> ### Author Response · Authors · 2022-11-18
> **Reply to reviewer s6p1**
>
> We thank you a lot for your review. We have uploaded a new version of the paper with changes based on your feedback. All essential changes are highlighted in blue color. We believe that the revision has significantly improved the quality of our paper. We hope those revisions resolve your concerns, thus improving your evaluation of our work and may make you consider raising the score. Below, we reply to your concerns point-by-point and summarize the related changes.
>
> > **Q1.** Main concern is the reliance on determining the sampling intervals when training a model. This seems like a critical hyperparameter and I don't have a sense for how easy it is to tune, and hence I'm not sure how easy it would be to extend the approach to other problem domains.
>
> We would like to first recall that in Table 3 (Appendix E) and Table 4 (Appendix F), we show the performance of controllers using different time grids. These results demonstrate that our algorithm is robust to the choice of the time grids in these two examples. So it is not hard to tune the hyperparameter in our examples. In the conclusion part, we recommend that at each iteration, one can compute the distance between the training data and data reached by the NN controller at different times (see Figure 2 for an example) and choose the time at which the distance starts to increase rapidly as the temporal grid for adaptive sampling. In our examples, this strategy performs well. We hope to make this process more systematic to make the choice of time grids easier in the future work.
>
>
> > **Q2.** The problem setup here involves applying control to a system over some finite interval $[0, T]$. How would this approach extend to a system that needs to operate continuously (e.g. a robot in a factory) if the premise here is that eventually control quality begins to degrade? Would you need to break up tasks into sub-tasks?
>
>
> This is an interesting question to consider. When the problem's horizon $T=\infty$, we may modify our algorithm according to the control's objective. If the goal is to operate continuously (e.g. a robot in a factory), as the reviewer suggests, it is natural to break up tasks into sub-tasks, each with reasonable finite horizons. If the goal is to stabilize the system for a long time, we know the states over the optimal paths will stay in a bounded domain after a certain time. If the neural-network-based controller is also stable, we know the states over the IVP paths will also stay in a bounded domain after a certain time. In this sense, we can view $[t_K, \infty)$ as an interval and apply the idea of IVP enhanced sampling prior to $t_K$ directly. We will leave the investigation of this direction to future work.
>
>
>
> > **Q3.** The problem formulation here assumes that we have knowledge of the system dynamics. What if those models are inaccurate? I would think that you would end up with a similar problem to the one you used to motivate this work, i.e. that the controls you apply bring you to states you haven't seen in your training data and that degrades performance more and more over time. Given that, do you think this approach could be extended to problems where dynamics models are inaccurate? I feel like this theme was somewhat explored in the experiments where noise was added to the dynamics, but it wasn't explicitly framed in this way.
>
>
> It is a very important question to handle optimal control problems with inaccurate models. The experiments with noise are our initial attempts to this problem. These experiments show that if the error of the model is small, we can still directly use the controller learned by the inaccurate model. However, when the error increases, the optimized controller's performance degrades quickly. If we can access the real dynamic through samples (like the setting in reinforcement learning), one possible approach is to compute all the IVPs under the simulation of the real dynamics and only use the inaccurate model to compute the open-loop optimal controllers. We expect this approach can further improve the performance and robustness of the learned closed-loop controller with respect to the inaccurate dynamics but this has not been systematically studied yet.
>
>
>
> > **Q4.** In the iterative training procedure do you use the same network throughout and finetune that network during each training stage or do you train a new network during each stage. It may have said this somewhere in the paper and I missed it, but it might be nice to state this more clearly e.g. within Algorithm 1.
>
> At each iteration, we train a new network from scratch. In the revision, we add this training detail in Appendix D and E. Which choice is more efficient for training, either training from scratch or finetuning the previous one, may depend on the specific problems. So we tend to treat this choice as a specific implementation detail. In our problems, we find that training a new network gives good performance more robustly.

---

### Official Review · Reviewer_9LKe · 2022-11-01

**Confidence:** 3
**Correctness:** 3
**Technical Novelty And Significance:** 2
**Empirical Novelty And Significance:** 2
**Recommendation:** 6

**Clarity, Quality, Novelty And Reproducibility:**

The paper is written clearly. The readability is good. However, the LQR setting for the theory study and the two examples for numerical study are relatively simple. It is quite difficult to see what is the unique novelty in the proposed approach. It is also unclear how to compare the proposed approach with policy-based approaches.


**Strength And Weaknesses:**

Strength:
1) The proposed approach seems to improve significantly over the vanilla approach on the two examples presented in this paper.
2) The proof for the theoretical results on LQR seems to be correct.


Weaknesses:
1) The numerical study is not sufficient to demonstrate the competitiveness of the proposed approach. How does the proposed approach compare with policy gradient (TRPO, PPO, SAC) on those problems?

 2) The theory is only provided for the simple LQR problem, and does not provide too much insights for nonlinear control problems, which are the main focus of this paper.

**Summary Of The Paper:**

This paper proposes the initial value problem enhanced sampling method to mitigate the distribution shift issue in applying supervised learning to solve the Hamilton-Jacobi-Bellman equation. The authors theoretically prove that the proposed sampling strategy improves over the vanilla strategy on the LQR problem by a factor proportional to the total time duration. Then they further numerically demonstrate that the proposed sampling strategy significantly improves the vanilla strategy on the optimal landing problem of a quadrotor and the optimal reaching problem of a 7 DoF manipulator.


**Summary Of The Review:**

I cannot recommend acceptance for this paper since the novelty is not very clear. The LQR setting for the theory study and the two examples for numerical study are relatively simple. It is also unclear how to compare the proposed approach with policy-based approaches.

========================

**Post rebuttal**

Thanks for addressing my comments. I increase my score to 6. However, I still think the theory in this paper is relatively weak. The paper will be much more insightful if the theory part can be strengthened.

---

> ### Author Response · Authors · 2022-11-18
> **Reply to reviewer 9LKe**
>
> We thank you a lot for your review. We have uploaded a new version of the paper with changes based on your feedback. All essential changes are highlighted in blue color. We believe that the revision has significantly improved the quality of our paper. We hope those revisions resolve your concerns, thus improving your evaluation of our work and may make you consider raising the score. Below, we reply to your concerns point-by-point and summarize the related changes.
>
> > **Q1.** The numerical study is not sufficient to demonstrate the competitiveness of the proposed approach. How does the proposed approach compare with policy gradient (TRPO, PPO, SAC) on those problems?
>
> Our algorithm is suitable for optimal control problems in which the system dynamics and objective is explicitly known for solving the problem. The typical policy gradient methods developed in the reinforcement learning community are designed for those problems in which we can only have samples of transition dynamics and rewards. Compared to reinforcement learning algorithms, optimal control algorithms often provide better results since they utilize the information from the explicit model of the dynamics. One typical example is that one can compute the exact gradient with an explicit dynamic model and cost, while in policy gradient algorithms, one can only estimate the gradient from samples.
>
> Back to our two specific examples, we have implemented the PPO algorithm using [Stable Baselines3]( https://github.com/DLR-RM/stable-baselines3). We consider the same settings in this paper and some easier settings. In all tested settings, the PPO algorithm either fails to converge or ends up with a cost much higher than optimal. In a simplified setting (much shorter traveling distance, simpler cost design and smaller action spaces) of the reaching problem of the manipulator, the best policy we can achieve still performs unsatisfactory with an average cost 20 times higher than the optimal one, after combining a few RL training techniques like reward engineering. It is worth mentioning that even the policy search approach (roughly speaking, replacing the estimated policy gradient with the exact gradient) still can not help find a reasonable policy. We remark that in typical applications of reinforcement learning algorithms in manipulators, the policy outputs the relative movement of the end-effector and requires another torque controller like a PD controller to drive motors to achieve desired joint positions. Our optimal reaching problem of the manipulator aims to find a policy that directly outputs torque, which is much more difficult than that setting due to the highly nonlinear dynamics.
>
>
> We believe that the comparison between the reinforcement learning algorithms and the optimal control algorithms is an important question. Nevertheless, in this paper, we mainly focus on the problems that can not be effectively handled by existing optimal control methods (policy search and supervised-learning-based approach). Therefore, the difficulty of the problems may not be suitable for the comparison between the reinforcement learning algorithms and the optimal control algorithms. To obtain more meaningful and systematic results, one should consider problems easier than that considered in this paper, which seems to be beyond the scope of this paper.
>
> > **Q2.** The theory is only provided for the simple LQR problem, and does not provide too much insights for nonlinear control problems, which are the main focus of this paper.
>
> Based on your and another reviewer's comment, we consider a more general setting in the LQR problem to allow the noise to have a non-zero expectation. Although these settings still can not reflect all the difficulties in the nonlinear control problems, we shall point out that in this LQR problem, the distribution mismatch phenomenon exists and the vanilla method can not obtain a satisfactory controller while the IVP enhanced sampling method significantly lessens the distribution mismatch phenomenon and provides a satisfactory controller, which is consistent with our motivation and the numerical results for general nonlinear control problems. Theoretical analysis beyond the LQR setting is important, and we hope we can cope with it in the future.

---

> > ### Comment · Reviewer_9LKe · 2022-11-26
> > **Thank you for your response.**
> >
> > Thanks for addressing my comments. The discussions on the numerical results are much more complete than the previous version. I increase my score to 6. However, I still think the theory in this paper is relatively weak. The paper will bring much more insights if the theory part can be strengthened.

---

### Author Response · Authors · 2022-11-18
**Revision highlight**

We would like to thank all reviewers for the comments and suggestions that helped improve the paper. In the updated version, we highlighted the main changes in blue color for ease of comparison. One main concern raised by the reviewers is some further comparison with other imitation learning algorithms like DAGGER (dataset aggregation) or reinforcement learning algorithms.

In the revision, we have added a comprehensive comparison between DAGGER and IVP enhanced sampling algorithm, in terms of concepts, theoretical results, and numerical results. In summary, the IVP enhanced sampling algorithm gives better performance consistently in different challenging problems.

We have also implemented the PPO algorithm using [Stable Baselines3]( https://github.com/DLR-RM/stable-baselines3) to solve our optimal landing problem and optimal reaching problem. We consider settings identical to our paper and some easier settings. In all tested settings, the PPO algorithm either fails to converge or ends up with a cost much higher than optimal (at best ~20 times higher than the optimal cost in a very simple setting of the reaching problem of the manipulator). We hypothesize that it is much more difficult for the RL algorithm to optimize a policy at the level of torque output, as in our problems, than optimizing a policy at the level of position displacement, as commonly done in the RL literature.

**These comparisons in terms of theoretical and numerical results highlight the superiority of the proposed algorithm in solving complex optimal control problems.**

---

### Decision · Program_Chairs · 2023-01-20

**Decision:**

Reject

**Justification For Why Not Higher Score:**

Reviewers feel that this work does not represent an advancement in the state of the art of control.

**Justification For Why Not Lower Score:**

N/A

**Metareview: Summary, Strengths And Weaknesses:**

This paper presents a closed-loop controller for non-linear dynamical systems. The key scientific idea is to formulate an initial value problem that mitigates distributional shift. The reviewers all agree that the ideas presented in the work, including the theoretical proofs are interesting (and correct). The additional discussions added to the paper further improve the readability of this paper.

Unfortunately, the reviewers also feel that this work does not represent an advancement in the state of the art of non linear control. Quoting a reviewer's discussion point: ' I still think the theory in this paper is too weak, and I agree with Reviewer s6p1 that this paper doesn't represent a major advancement over the state of the art.' Since one of the claims of this work is that it provides improved non-linear control, the lack of seeds and error bars in the results are also concerning.

Overall, I believe this paper is almost there. With a few tweaks and improvements in the experimental evaluation, I believe this work should be resubmitted to an upcoming conference.